# Future Arctic ozone recovery: the importance of chemistry and dynamics

Ewa M. Bednarz[1], Amanda C. Maycock[1,2,#], N. Luke Abraham[1,2], Peter Braesicke[1,2,*], Olivier Dessens[3], John A. Pyle[1,2]

[1] Department of Chemistry, University of Cambridge, Cambridge, UK
[2] National Centre for Atmospheric Science - Climate, UK
[3] University College London, London, UK
[#] now at: School of Earth and Environment, University of Leeds, Leeds, UK
[*] now at: Karlsruhe Institute of Technology, Institute for Meteorology and Climate Research, Karlsruhe, Germany

*Correspondence to*: E. M. Bednarz (emb66@cam.ac.uk)

**Abstract.** Future trends in Arctic springtime total column ozone, and its chemical and dynamical drivers, are assessed using a 7 member ensemble from the Met Office Unified Model with United Kingdom Chemistry and Aerosols (UM-UKCA) simulating the period 1960-2100. The Arctic mean March total column ozone increases throughout the 21$^{st}$ century at a rate of ~11.5 DU decade$^{-1}$, and is projected to return to the 1980 level in the late 2030s. However, the integrations show that even past 2060 springtime Arctic ozone can episodically drop by ~50-100 DU below the corresponding long-term ensemble mean for that period, reaching values characteristic of the near present day average level. Consistent with the global decline in inorganic chlorine (Cl$_y$) over the century, the estimated mean halogen induced chemical ozone loss in the Arctic lower atmosphere in spring decreases by around a factor of two between 1981-2000 and 2061-2080. However, in the presence of a cold and strong polar vortex elevated halogen induced ozone losses well above the corresponding long-term mean continue to occur in the simulations into the second part of the century. The ensemble shows a significant cooling trend in the Arctic winter mid- and upper stratosphere, but there is less confidence in the projected temperature trends in the lower stratosphere (100-50 hPa). This is partly due to an increase in downwelling over the Arctic polar cap in winter, which increases transport of ozone into the polar region as well as drives adiabatic warming that partly offsets the radiatively-driven stratospheric cooling. However, individual winters characterised by significantly suppressed downwelling, reduced transport and anomalously low temperatures continue to occur in the future. We conclude that despite the projected long-term recovery of Arctic ozone, the large interannual dynamical variability is expected to continue in the future, thereby facilitating episodic reductions in springtime ozone columns. Whilst our results suggest that the relative role of dynamical processes for determining Arctic springtime ozone will increase in the future, halogen chemistry will remain a smaller but non-negligible contributor for many decades to come.

# 1 Introduction

There was a period of rapid growth in the use of chlorofluorocarbons starting from the 1960s; it is now well understood that these compounds, essentially inert in the troposphere, undergo photo-degradation in the stratosphere and that their breakdown products initiate depletion of the stratospheric ozone layer (Molina and Rowland, 1974). In 1985, Farman et al.
reported a significant springtime depletion of ozone in Antarctica. Statistical analyses of long-term ground-based and satellite data sets subsequently demonstrated widespread, albeit smaller, downward trends at many locations (see, e.g., the Report of the International Ozone Trends Panel, WMO, 1988). These discoveries prompted a massive international research effort, which later confirmed the key role of halogen species in driving both the polar (Anderson et. al., 1989) and mid-latitude ozone losses (Hadjinicolaou et. al. 1997). Based on this scientific evidence, an international treaty, the Montreal
Protocol on Substances that Deplete the Ozone Layer, came into force in 1987, and subsequent amendments and adjustments have significantly strengthened the regulatory control. Accordingly the stratospheric abundances of both chlorine and bromine peaked around the turn of the century and are now falling slowly, consistent with the long atmospheric lifetimes of some of the main ozone depleting substances (ODS) (SPARC, 2013; WMO, 2014).

While the major research questions were initially around the processes leading to ozone loss, the agenda has now shifted to
the question of how and by when ozone levels in the stratosphere will return to earlier, historical values. However, it has been clear for some time (see Hofmann and Pyle, Ch. 12, WMO/UNEP, 1999) that ozone will not simply follow a path to recovery that is symmetric about the peak halogen loading; continuing increases in greenhouse gas (GHG) abundances mean that a future low-halogen stratosphere will not be the same as in the past.

Recent assessments of the state of the ozone layer (WMO, 2011, 2014; see also Eyring et al., 2010) paint a consistent picture
of possible trajectories of ozone recovery based on integrations with chemistry-climate models (CCMs). First, because of the cooling of the stratosphere by GHGs (mainly $CO_2$), recovery of ozone to a particular level will occur before the corresponding recovery of halogen levels. Secondly, mid-latitude recovery in either hemisphere will occur before polar recovery. Recovery in the Arctic is expected by about 2030, but with a large inter-model range (see, e.g., Fig. 3.11 of WMO, 2011). Modelled recovery of Antarctic springtime ozone occurs on average about 25 years later.

Recovery of Arctic ozone is particularly interesting and is the focus of this study. While similar chemical processes operate in the Arctic as in the Antarctic, where substantial springtime loss occurs every year, meteorological conditions in the Arctic are generally less favourable to cause extreme ozone depletion (Solomon et al., 2007, 2014). In particular, the Arctic stratospheric polar vortex in wintertime is, on average, more dynamically disturbed, resulting in higher temperatures and greater transport of ozone into the polar regions, and consequently the Arctic is generally not subject to the very large
springtime ozone losses observed in the Antarctic (Tilmes et. al., 2006, Solomon et. al., 2007, 2014). However, when the winter/spring Arctic lower stratosphere is cold for a long period, substantial ozone depletion is to be expected (WMO, 2014). Occurrences of significant chemical depletion have been reported for a number of Arctic winters in the last two decades (e.g. Goutail et al., 1999; Harris et al., 2002; Rex et al., 2002; Tilmes et al., 2004; Rex et al. 2006; Kuttippurath et al., 2010), and

these have the potential to significantly impact on ecosystems and human populations. Indeed, the reduction of Arctic column ozone observed in late winter and early spring of 2011 was as large as that observed in the Antarctic (e.g. Manney et al., 2011). (Note however that Arctic column ozone is generally much higher than in the Antarctic, so that the late March values in 2011 were about twice those seen typically in springtime in the south, despite a comparable level of depletion).

The evolution of Arctic ozone will depend not only on the future trends in GHGs and ODSs but also on the behaviour of the stratospheric polar vortex, which is highly variable from year-to-year. In general, chemical and dynamical drivers of polar ozone depletion are likely to be related on interannual timescales (Tegtmeier et al., 2008). A strong and cold polar vortex favours low ozone. Transport of ozone to the high latitudes is much reduced and low temperatures in the lower stratosphere promote the formation of polar stratospheric clouds (PSCs) and subsequent chlorine activation, thereby enhancing chemical

ozone loss. In addition to PSCs, chlorine activation can also occur on cold sulphate aerosols (e.g. Hanson et al., 1994; Drdla and Müller, 2012; Wegner et al., 2012; Solomon et al., 2015). In contrast, a warmer, more disturbed polar vortex should be accompanied by much smaller chemical destruction of ozone and enhanced transport from lower latitudes. The large interannual variability seen in Arctic late winter/early spring total column ozone results from complex interactions between these chemical and dynamical processes. It has been estimated that these two factors have contributed in roughly equal

measure to recent variations in late winter/early spring total ozone (Rex et al. 2004, Tegtmeier et al., 2008).

There has been keen debate about the future evolution of conditions in the Arctic winter stratosphere. The stratosphere is expected to cool in the global mean under increased GHG concentrations (e.g. Fels et al., 1980), but the evolution of the highly variable Arctic lower stratosphere cannot be predicted with confidence. Rex et al. (2004) have reported a strong linear correlation between observed wintertime chemical ozone loss and a measure of the volume of PSCs ($V_{PSC}$). It has been

suggested that long-term temperature changes in the Arctic polar lower stratosphere may have contributed to the coldest Arctic winters becoming significantly colder in recent decades (Rex et al., 2004; Rex et al., 2006; Ivy et al., 2014), reflecting the conditions that lead to larger $V_{PSC}$ and facilitate greater chemical ozone losses. In contrast, Rieder and Polvani (2013) showed that increases in $V_{PSC}$ since 1979, as estimated from multiple reanalysis datasets, are not highly statistically significant. While the future evolution of the Arctic polar vortex is the subject of keen debate, it is clear that any changes in

the future will have a substantial influence on the Arctic ozone column.

Langematz et al. (2014) have studied the future evolution of Arctic ozone and temperature using a CCM. They found that rising GHG concentrations lead to a cooling of the Arctic lower stratosphere in early winter, but that there were no significant temperature changes in late winter or spring. They did not find a long-term downward trend in minimum Arctic temperatures, nor any extension of the vortex break-up date into later spring. Consistent with numerous other studies (e.g.

WMO, 2011, 2014), they found that Arctic ozone is expected to increase in the latter part of the century.

Langematz et al. (2014) examined two transient 21[st] century integrations (with and without GHG changes) and a number of timeslice experiments with different boundary conditions. Our focus on Arctic ozone is similar to theirs, but we employ a different approach. We examine the evolution of Arctic springtime ozone in the UM-UKCA CCM (Morgenstern et al., 2009) using an ensemble of 7 transient simulations carried out as part of the World Climate Research Programme (WCRP)

Stratosphere-troposphere Processes and their Role in Climate (SPARC) chemistry-climate model initiative (CCMI, Eyring et al., 2013). This ensemble of simulations enables us to explore the year-to-year variability in Arctic polar ozone and its relation to the long-term trends in chemical and dynamical drivers over the 21$^{st}$ century. Section 2 gives information on the model and simulations performed. Section 3.1 describes the temporal evolution of Arctic springtime total column ozone. Section 3.2 examines the chemical, radiative and dynamical drivers of Arctic ozone during the 21$^{st}$ century. Section 3.3 highlights the importance of these drivers in a case study for individual low and average ozone events simulated in the second half of the 21$^{st}$ century. Finally, Section 4 summarises the key results.

## 2 Model, experiment and methods

### 2.1 The Model

We use the UM-UKCA CCM that is built around the Met Office Unified Model (MetUM) in the HadGEM3-A configuration (Hewitt et al., 2011) at MetUM version 7.3. The model uses a horizontal resolution of 2.5° latitude by 3.75° longitude, with 60 vertical levels up to 84 km. We use the extended Chemistry of the Stratosphere (CheS+) chemistry scheme, which is an expansion of Morgenstern et al. (2009) where CFC-11, CFC-12, CFC-113, HCFC-22, Halon-1211, Halon-1301, $CH_3Br$, $CH_3Cl$, $CCl_4$, $CH_2Br_2$, and $CHBr_3$ are considered explicitly, resulting in an additional 17 bimolecular and 9 photolytic reactions. The chemical tracers $O_3$, $CH_4$, $N_2O$, CFC-11, CFC-12, CFC-113, and HCFC-22, are all interactive with the radiation scheme. The model and chemistry scheme were used for the recent SPARC Report on the Lifetimes of Stratospheric Ozone-Depleting Substances, Their Replacements, and Related Species (SPARC, 2013; Chipperfield et al., 2014).

As in Morgenstern et al. (2009), heterogeneous reactions on PSCs as well as formation and removal of nitric acid trihydrate (NAT) PSCs follow Chipperfield (1999), with the formation and removal of ice PSCs included in the hydrological cycle. Regarding the NAT PSCs, as described in Chipperfield et al. (1999), the NAT PSC formation follows the equilibrium expression from Hanson and Mauersberger (1988). We note that this is a simple approach; in reality the formation of NAT particles requires supersaturation of $HNO_3$ over NAT to occur (see e.g. Solomon et al., 2015). The denitrification scheme assumes a relatively slow NAT sedimentation velocity of ~40 m/day for pure NAT PSCs, and a much faster NAT sedimentation velocity of ~1540 m/day in the presence of ice; the latter assumes coating of NAT PSCs onto ice particles. The reactive uptake coefficients for the heterogeneous chemical reactions on NAT/ICE PSCs used in the model are listed in Table S1 of the Supplementary Information. Stratospheric aerosols are prescribed for heterogeneous reactions, as well as in the photolysis and radiation schemes. Note that unlike on PSCs, the heterogeneous reactions between $ClONO_2$ and HCl as well as between $N_2O_5$ and HCl on sulphate aerosols are not included in the scheme. As heterogeneous reactions on liquid aerosols can be important for the springtime ozone (see e.g. Solomon et al., 2015), the model heterogeneous chemistry scheme does not represent an exhaustive and fully realistic treatment of heterogeneous processes. However, using a similar model version Keeble et al. (2014) obtained a reasonable representation of the springtime Antarctic ozone hole, with the

majority of the simulated Antarctic ozone depletion in the model driven by the heterogeneous reactions on the surfaces of NAT and ice PSCs.

The 11-year solar cycle variability is included consistently in both the radiation and photolysis schemes. In the radiation scheme, we use the method employed in HadGEM1 (Stott et al., 2006) and HadGEM2-ES models (Jones et al., 2011). Total

solar irradiance (TSI) data are those recommended in the fifth Coupled Model Intercomparison Project (CMIP5) (Wang et al., 2005; Lean, 2009), processed to force the mean of the 1700-2004 period to be 1365 W/m$^2$ (Jones et al., 2011). A fit to spectral data from Lean (1995) is used to account for a change in partitioning of solar radiation into wavelength bins. In the Fast-JX photolysis scheme (Telford et al., 2013), the change in partitioning of solar irradiance into wavelength bins is accounted for by scaling the bins according to the CMIP5 spectral solar irradiance data (Wang et al., 2005; Lean, 2009) for

the years 1981 and 1986, and the long-term evolution of the processed TSI timeseries. Solar cycle variability for the future period (after 2009) is included as in earlier periods but with a repeating sinusoidal 11-year cycle with an amplitude derived from observed cycle 23 (see Jones et al., 2011; Gray et al., 2013).

The stratospheric climatology and variability, including that in the Arctic region, have been evaluated in similar stratosphere-

resolving versions of the MetUM (Osprey et al., 2010; Hardiman et al., 2010) and UM-UKCA (Morgenstern et. al. 2009). The model includes parameterized orographic and non-orographic gravity wave drag (Scaife et al., 2002; Webster et al., 2003), and simulates an internally generated quasi-biennial oscillation (QBO) (Scaife et al, 2002). As shown by Morgenstern et al. (2009), the model reproduces the observed anti-correlation between polar ozone and jet strength in March, which is one indicator of the coupling between chemistry and meteorology discussed above.

A comparison between the Arctic/Antarctic mean (65-90°N/S) total ozone columns in a similar model version as used here, but with a specified dynamics set-up, i.e. in which model's temperature and wind fields are nudged towards meteorological reanalysis data, shows a good correlation between simulated and observed polar column ozone over the recent past (R=0.88 and R=0.96 for the Arctic and Antarctic mean, respectively, see Supplementary Material Fig. S1). In the Antarctic, there is some overestimation of the ozone column in the model by up to ~15-30 DU (see Fig. S1). For the cold March 2011, the

model shows a positive bias in the zonal mean monthly mean ozone levels in the Arctic lower stratosphere (30-100 hPa) of up to ~0.9 ppb (~41%; for the 70-90°N mean, 50 hPa) compared with MIPAS satellite data (Fisher et al., 2008, not shown). The positive ozone bias is commensurate with a negative bias in the zonal mean monthly mean ClO levels at this altitude. Note that there is some uncertainty in the observed zonal mean monthly mean ClO, in part due to sparse temporal sampling (twice a day) of the MIPAS instrument. In general, all models will show some biases with respect to observations.

Importantly, as in many studies, we compare the model for the present day and future periods in an internally consistent way; therefore, any biases will become less relevant for our study.

The mean age of air in the tropical lower stratosphere, as well as in the northern high-latitudes, is at the young end of observationally-derived values, but its gradient between the northern mid-latitudes and the tropics is in a fair agreement with observations and other models (SPARC, 2013; Chipperfield et al., 2014). This indicates a significant improvement in its

circulation as compared with the older atmospheric model version presented in Morgenstern et al. (2009) and in SPARC Report No.5 (SPARC, 2010).

## 2.2 The CCMI REFC2 Experiment

We follow the experimental design of the CCMI REF-C2 experiment (Eyring et al., 2013). Lower boundary conditions are

used for $CH_4$, $N_2O$, $H_2$, CFC-11, CFC-12, CFC-113, HCFC-22, Halon-1211, Halon-1301, $CH_3Br$, $CH_3Cl$, $CCl_4$, $CH_2Br_2$, $CHBr_3$, and $CH_3CCl_3$, with values for ODSs specified from WMO (2011) and future GHG abundances are specified according to the representative concentration pathway RCP6.0 (Fujino et al., 2006; Hijioka et al., 2008; Taylor et al., 2012). The prescribed stratospheric aerosols are as given by the SPARC climatology (SPARC, 2006). Sea-surface temperatures (SSTs) and sea-ice concentrations (SICs) are taken from one ensemble member (r2i1p1) of the HadGEM2-ES RCP6.0

ensemble (MOHC, 2011; Jones et al., 2011). We perform two integrations from 1960-2099 and five over a shorter period from November 1980 to December 2080; the latter five were initiated from different initial conditions taken from a supporting perpetual year 1980 integration. As a result of a data issue, a total of five 6-year-long periods is excluded from the analysis of the ensemble (see Supplementary Information for more details). When compared against the ERA-Interim reanalysis (Dee et al., 2011), the integrations show a somewhat weaker and warmer present day Arctic stratospheric vortex

in early/mid-winter, with a slight zonal wind bias of up to ~6 ms$^{-1}$ in the mid-latitude lower/mid- stratosphere in March (not shown).

## 2.3 Diagnostics of chemical loss

We use two diagnostics to estimate the chemical ozone loss in the Arctic region. Firstly, we estimate the ozone loss due to halogen reactions directly from reaction fluxes. We use the diagnostic framework of Lee et al. (2002) in which the rate of

odd oxygen ($O_x=O_3+O(^3P)+O(^1D)$) destruction is estimated for different catalytic cycles by determining the rates of their rate-limiting steps. The most important halogen-catalysed ozone loss cycles in the polar lower stratosphere are the ClO dimer cycle (Cycle 1, Molina and Molina, 1987), ClO+BrO cycle (Cycle 2, Yung et al., 1980; McElroy et al., 1986) and the more minor ClO+O($^3P$) cycle (Cycle 3, Stolarski and Cicerone, 1974):

$$ClO + ClO + M \rightarrow Cl_2O_2 + M$$
$$\mathbf{Cl_2O_2 + hv \rightarrow Cl + ClOO}$$
$$ClOO + M \rightarrow Cl + O_2 + M$$
$$\underline{2* (Cl + O_3 \rightarrow ClO + O_2)}$$
$$(NET: 2*O_3 \rightarrow 3*O_2 \; ; \; Cycle\ 1)$$

$$\mathbf{BrO + ClO \rightarrow Br + ClOO}$$          $$\mathbf{BrO + ClO \rightarrow BrCl + O_2}$$
$$ClOO + M \rightarrow Cl + O_2 + M$$          $$BrCl + hv \rightarrow Br + Cl$$

$$Br + O_3 \rightarrow BrO + O_2$$
$$\underline{Cl + O_3 \rightarrow ClO + O_2}$$
( NET: $2*O_3 \rightarrow 3*O_2$ ; *Cycle 2*)

$$\mathbf{ClO + O(^3P) \rightarrow Cl + O_2}$$
$$\underline{Cl + O_3 \rightarrow ClO + O_2}$$
( NET: $O_3 + O \rightarrow 2*O_2$ ; *Cycle 3*)

The rate-limiting step in each cycle is highlighted in bold font. Three further halogen cycles of lesser importance in the winter polar lower stratosphere (the BrO + O($^3$P) cycle analogous to Cycle 3, and a pair of ClO + HO$_2$ and BrO + HO$_2$ cycles) are also included. In all six reactions, a net loss of 2 odd oxygen molecules occurs per cycle. Assuming that $[O_x] \approx [O_3]$ and, thus, $d[O_x]/dt \approx d[O_3]/dt$, we then calculate the rate of ozone loss from each cycle and integrate in time (from 1 November to 30 March) and altitude (from the surface to 25 km), resulting in an estimate of the cumulative ozone loss at each grid point due to these halogen reactions in the polar lower stratosphere. Whilst the rates (i.e. fluxes) of ozone loss are calculated at each grid point and every UM-UKCA timestep, we use the zonal mean monthly-mean diagnostics for computational ease (except for vortex-average quantities reported in Sect. 3.3, where daily means are used). The cut-off altitude of 25 km was chosen so as to capture most of the region subject to halogen activation on PSC surfaces. Since no further separation is made between the ozone losses due to heterogeneous and homogenous chemistry, the diagnostic is not equivalent to the ozone losses initiated exclusively by the heterogeneous reactions on PSCs, as presented in e.g. Chipperfield and Jones (1999), Chipperfield et al. (2003) and Keeble et al. (2014).

It is important to understand the characteristics of this diagnostic when integrated over the winter. First, we have calculated losses in an Eulerian framework and so our values are not directly comparable to the Lagrangian estimates that have been calculated during some polar winters (see Rex et al. (2002) and Harris et al. (2002) for detailed comparison). Second, our values are integrated over 24 hours and will necessarily be lower than instantaneous rates calculated, for example, from observed, daytime observations of ClO. Third, unless otherwise specified we present data that have been averaged from 65°N to the pole. Unlike in the Antarctic, the Arctic vortex is not usually centred on the pole; it is mobile and highly variable, so our diagnostic will include areas where polar halogen chemistry is not active. However, the strength of our diagnostic is that it allows us to compare different model winters against each other. By definition, greater values of the diagnosed loss will be found in those winters where halogen chemistry is more important.

An additional diagnostic for ozone loss used in Sect. 3.3 is a passive ozone tracer, similar to that implemented in Chipperfield and Jones (1999) and Chipperfield et al. (2003). This chemically-inert tracer is initialised to the modelled ozone concentrations on 1 November each year. Whilst undergoing no chemical production or loss, it is transported by the circulation until the end of March. The difference between the chemical ozone field and the passive ozone tracer at the end

of each winter represents the change in ozone levels due to all chemical processes. This diagnostic was only included in one ensemble member, discussed in Sect. 3.3.

## 3 Results

### 3.1 Long-term evolution of polar total column ozone and $Cl_y$

Figure 1(a) shows timeseries of March total column ozone averaged over 65-90°N (henceforth referred to as Arctic mean) for the 7 UM-UKCA ensemble members. An analogous plot for 65-90°S in October is shown in Fig. 1(b). The corresponding Arctic mean total inorganic chlorine level ($Cl_y$) at 20 km is shown in Fig. 1(c).

The results in Fig. 1 are consistent with the multimodel means reported in WMO (2011, 2014) and Eyring et al. (2010). There is a reduction in Arctic total column ozone from the 1960s to the late 1990s, with the March ensemble mean dropping by ~15% to ~400 DU. This is consistent with the modelled increase in $Cl_y$ over this period. A similar long-term decline is modelled in the springtime Antarctic, where the ozone reduction is much larger in both absolute and percentage terms (125 DU change in October, i.e. ~38%). This decline in 20[th] century polar ozone agrees well with observations (red curve; Bodeker total ozone column data set, v2.8, Bodeker et al., 2005; Müller et al., 2008).

As a result of the Montreal Protocol and its subsequent amendments and adjustments, the increase in stratospheric $Cl_y$ abundances has ceased and values are projected to decrease over the 21[st] century, returning to their 1980 levels by about 2060 in both the Antarctic and Arctic (WMO, 2011). In the Antarctic, the average total ozone returns to 1980 levels over approximately the same period and therefore closely follows the evolution of $Cl_y$ (not shown). In contrast, total column ozone in the Arctic returns to 1980 levels by the late 2030s, which is approximately 15-20 years earlier than the return date for $Cl_y$ (Fig. 1(c)). In the Antarctic, the polar vortex is stronger climatologically and the evolution of ozone is largely determined by changes in halogen chemistry, in agreement with Austin and Wilson (2006). In the Arctic, the return date in UM-UKCA is somewhat later than indicated by the multi-model mean from the Chemistry Climate Model Validation 2 (CCMVal-2) project (Eyring et al., 2010), but is within the range of individual model estimates (WMO, 2011).

The individual curves in Fig. 1(a) and 1(b) highlight that the interannual variability in springtime total column ozone is much larger in the Arctic than in the Antarctic, with modelled values ranging from 346-487 DU in the 1981-2000 period. This large variability remains throughout the 21[st] century in the presence of the gradual reduction in $Cl_y$ and the long-term increase in mean Arctic ozone. The ensemble mean ozone increases by about 11.5 (±1.3, i.e. ±2 standard errors) DU decade[-1] over the 2000-2080 period, with its 11-year running mean increasing from 408 DU in 2000 to 482 DU in 2074. The minimum monthly mean ozone levels at a single location anywhere poleward of 65°N increase at a similar rate to the Arctic mean ozone column, albeit the 11-year running mean value of this quantity in 2000 is ~50 DU lower than the corresponding Arctic mean value. The persistence of the large interannual variability is associated with years with particularly low Arctic ozone throughout the 21[st] century. In the second half of the century, March column ozone episodically drops not only below

the 1980 level, but also to values of ~410-415 DU, which are close to the long-term minimum of the 11-year running ensemble mean around the turn of the century. In addition, one ensemble member simulates 383 DU in March of 2063 (see green point in Fig. 1(a)), which is comparable to the average values routinely found under present day conditions.

Figure 2 shows probability density functions (PDFs) of Arctic mean March total ozone for five 20-year intervals from 1981 to 2080. As is also evident in Fig. 1(a), the means of the PDFs (coloured diamonds in Fig. 2) progressively increase over the $21^{st}$ century, consistent with the gradual decline in stratospheric $Cl_y$ (coloured points in Fig. 2) and the super recovery of ozone (Eyring et al., 2010). Note that the first two decades of the $21^{st}$ century have, on average, lower ozone compared to 1981-2000 (~14 DU decrease in mode, Kolmogorov-Smirnov (KS) test p-value of 0.052), with a suggestion of a change in skewness of the distribution. This results from the rate of reduction in column ozone at the end of the $20^{th}$ century being larger than the rate of the subsequent recovery, which begins near the turn of the century.

Notably, the distributions of Arctic March column ozone are negatively skewed, which indicates the occurrence of ozone values well below the 20-year mean. Austin and Wilson (2006) reported an increase in the interannual variability of March Arctic column ozone in the future. Clearly, the precise characteristics of the PDFs in Fig. 2, such as their width and skewness, vary between the five periods shown. Although there could in principle be a forced trend in these characteristics with time (e.g. Rex et al., 2004; Austin and Wilson, 2006), such factors will also be influenced by internal variability and decadal variability in SSTs/SICs and solar forcing, which are common to all of the simulations (see Sect. 2). With the exception of an increase in the mean and mode of ozone over the century, we do not find systematic changes in other characteristics of the PDFs, such as their width, with time.

The occurrence of low ozone episodes in the future, with values episodically dropping by up to ~50-100 DU below the 11-year running mean, is particularly clear in the PDF for the last two decades modelled (2061-2080). In a single transient integration, Langematz et al. (2014) found that springtime Arctic ozone did not generally drop below the 1970-82 mean beyond around 2060. In the UM-UKCA simulations, years in which the springtime Arctic ozone drops below the 1980 level (~450 DU, see Fig. 1(a)) continue until at least 2080. This contrasting result may be due to differences, likely dynamical, in the representation of the Arctic winter stratosphere and its variability between the CCMs used in the studies, differences in experimental set-up (e.g. GHGs, SSTs and SICs), or the fact that we have more fully sampled the internal variability of the climate system by using an ensemble of integrations.

### 3.2 Future Arctic variability and trends in the chemical and dynamical drivers of ozone

Polar ozone is influenced by a complex interplay between chemical and dynamical processes. Over the $21^{st}$ century, there may be forced trends in factors that affect both sets of processes, including atmospheric abundances of key chemical species, stratospheric temperatures, as well as the large-scale circulation; all of these may contribute to the evolution of ozone during this century. The following section discusses the temporal evolution of some of the chemical and dynamical processes that affect Arctic ozone in the UM-UKCA ensemble.

### 3.2.1 Trends and variability in chemical drivers

An estimate of the contribution of halogen induced chemical loss to the Arctic mean March column ozone and its time dependence is shown in Fig. 3(a). The scatterplot shows the Arctic mean March total column ozone versus the cumulative winter-time (1 November to 30 March) chemical ozone loss due to the six halogen cycles of most importance in the polar

lower stratosphere integrated from the surface to 25 km, calculated using the Eulerian diagnostics discussed in Sect. 2.3. Note that the diagnostic is henceforth referred to interchangeably as 'halogen induced ozone loss' and, in short, 'halogen loss'. We use the diagnostic to compare particular Arctic winters against each other. The diagnostic constitutes a useful tool for examining both the interannual and interdecadal variability in the halogen induced ozone loss in our model and its contribution to variability in ozone. Recall that the calculated loss should not be expected to match quantitatively Lagrangian

calculations of the chemical ozone loss in particular Arctic winters (e.g. Harris et al., 2002; Rex et al., 2004; Tegtmeier et al., 2008).

There is a significant correlation between the March total column ozone and the cumulative halogen induced loss during winter (Pearson correlation coefficient of -0.81 for the full distributions of all points). Around half of the estimated halogen induced ozone loss can be attributed to the ClO+BrO cycle, with further contributions from the ClO dimer and the

ClO+O($^3$P) cycles; the remaining three cycles together contribute a background of ~5-7 DU that varies only very little on both short and long-term timescales (not shown). As expected, there is a gradual decline in the average halogen induced loss over the 21$^{st}$ century, with the mean diagnosed loss almost halving between 2001-2020 and 2061-2080 (36 DU and 20 DU, respectively). This reflects the long-term decrease in the global $Cl_y$ levels, which also decline by about a factor of two in that period (Fig. 1(c) and 2).

Despite the long-term decrease in the mean halogen induced ozone loss, even after 2060 the halogen loss in individual years can still be more than a factor of two higher (i.e. by ~20-30 DU) than the 11-year running mean for that period (Fig. 3(b)). For example, the largest chemical losses during 2061-2080 are greater than 40 DU; this illustrates the continued potential for enhanced chemical loss to occur in the Arctic in the presence of favourable dynamical conditions (i.e. a cold and strong vortex). So, although the maximum halogen induced ozone losses later in the century are lower, as expected, than found

during the period between 1980 and 2040 (Fig. 3(a)), these processes still make an important contribution to the overall ozone anomaly in these years.

Chipperfield and Jones (1999) estimated the contribution of ozone losses due to heterogeneous halogen chemistry to the interannual variability in Arctic springtime ozone by comparing the standard deviations of diagnosed chemical loss and March total ozone column. In our calculations, after removing the 11-year running ensemble mean from both the Arctic

column ozone and halogen induced ozone loss time series, we find that the standard deviation of the estimated halogen loss constitutes nearly 30% of the standard deviation of the modelled March total column ozone over the entire 1981-2080 period of the integration. So, while halogen losses clearly contribute to the springtime Arctic ozone, the dominant driver in the model that determines the interannual variability is dynamics, as in the study of Chipperfied and Jones (1999).

### 3.2.2 Trends in Arctic stratospheric temperatures

There is considerable interest in whether there has been any trend in minimum temperatures in the Arctic winter lower stratosphere in recent decades. This could increase the amount of PSCs formed in winter as well as their persistence, thus enhancing chemical ozone loss; such an increase, if present, could have contributed to years with low springtime ozone observed in the recent past (see e.g. Rex et al., 2004; Rieder and Polvani, 2013; Ivy et al., 2014). The direct radiative impact of increasing $CO_2$ levels is to cool the stratosphere, with the greatest impact in the upper stratosphere (Fels et al., 1980). Temperatures in the Arctic winter stratosphere are also strongly influenced by dynamical processes, which can enhance or offset the radiatively driven cooling (e.g. Bell et. al., 2010; Butchart et al., 2010; Langematz et al., 2014). In addition to long-term changes, the interannual variability is high and can potentially compound the identification of trends on shorter timescales.

Figures 4(a-c) show simulated Arctic mean temperature trends [K decade$^{-1}$] at five pressure levels in the stratosphere for the period 1981-2080 in early (November-December), mid (January-February) and late winter/spring (March), respectively. Black points denote the trends calculated for individual ensemble members, with the trend for the ensemble mean shown in red along with ±2 standard errors. A statistically significant cooling trend is found in the mid and upper stratosphere (at and above 30 hPa) throughout the winter. This is most robust in early winter (Fig. 4(a)) and late winter/spring (Fig. 4(c)), when all ensemble members show trends of the same sign. The magnitude of the trend increases with decreasing pressure in agreement with earlier studies (Fels et al., 1980; Bell et. al. 2010, Oberländer et al., 2013; Langematz et al., 2014). However, in the lower stratosphere (100-50 hPa) there is less confidence in the projected temperature trends across the ensemble throughout the winter. In early winter, the ensemble mean shows a weak cooling in the Arctic lower stratosphere (-0.15 K decade$^{-1}$ at 50 hPa; trend small and not statistically significant at 100 hPa). At least one ensemble member shows a near zero trend. In comparison, Langematz et al. (2014) found a statistically significant cooling trend in early winter over 1960-2100 throughout the polar stratosphere. In late winter/early spring, no trend in temperature was found in our study at 100 hPa. At 50 hPa and above, as discussed above, the results suggest an overall cooling trend; however, a large spread of magnitudes can be seen across the ensemble, with individual 50 hPa March temperature trends ranging from ~0 to ~0.5 K decade$^{-1}$. In mid-winter, the intra-ensemble spread is even larger and no significant trend in the ensemble mean lower stratospheric temperature is found.

A similar analysis of future trends in the local minimum Arctic temperature anywhere poleward of 65°N ($T_{min}$) calculated from monthly mean data reveals largely qualitatively similar results to the Arctic mean quantities shown in Fig. 4 (not shown). The results support the need for ensemble studies to confidently detect simulated temperature trends in the polar lower stratosphere.

### 3.2.3 Trends in $V_{PSC}$

Stratospheric $H_2O$ and $HNO_3$ levels are projected to increase in the future, which is likely to enhance levels of PSCs (Langematz et al., 2014). Moreover, the formation of PSCs will be further strengthened in winters with colder polar temperatures. Figure 5 shows a scatterplot of November to March mean potential $V_{PSC}$ versus the halogen induced ozone loss

in the polar lower stratosphere, as discussed in Sect. 3.2.1. Potential $V_{PSC}$ is defined as average daily mean volume of air (1. November to 30. March, 1-30 km) in which NAT PSCs are thermodynamically possible according to the NAT equilibrium scheme used in the model. There is a tendency in the model for higher potential $V_{PSC}$ to be associated with higher halogen induced ozone losses ($R \approx 0.8$ for the individual 20 year periods in the 21[st] century), in broad agreement with the observed relationship (Rex et al., 2004, 2006).

For the 1993-2005 period (as studied in Rex et al., 2006), the gradient of the linear fit (R=0.81) is 1.03 ($\pm$0.17) $DU/10^6$ $km^3$, which is about half of the value reported by Rex et al. (2006). We caution against a detailed quantitative comparison because of: differences between the Eulerian and Langrangian halogen-induced ozone loss diagnostics (see Sect. 2.3); the use of 65-90ºN average quantities instead of vortex averages; differences in the altitudes over which chemical loss has been integrated; as well as differences in the definitions of $V_{PSC}$.

The amount of halogen induced ozone loss past ~2040 that occurs for a given potential $V_{PSC}$ is reduced with respect to the earlier periods, consistent with the projected future reduction in $Cl_y$ levels and the resulting lower chlorine levels available for activation in the future period. However, the period 2061-2080 is characterised by occurrences of particularly high potential $V_{PSC}$; this is consistent with the occurrence of a number of very cold Arctic winters in the simulations (see Sect. 3.2.4 for the discussion of variability in the residual circulation), and could also be related to the rising $HNO_3$ and $H_2O$ levels

(not shown, see also Langematz et al., 2014). Hence, despite the lower $Cl_y$ in the future, higher potential $V_{PSC}$ still increases the likelihood for chlorine activation. Even though dynamics and transport will dominate the interannual variability of Arctic ozone in the future, halogen chemistry will still have the potential to contribute to instances of low Arctic springtime total column ozone.

### 3.2.4 Trends in atmospheric circulation and transport

In contrast to the Antarctic region, where interannual and interdecadal variability is smaller and, thus, springtime ozone levels within the polar vortex are mostly determined by the amount of halogen-induced loss, springtime ozone in the Arctic is strongly influenced by variability in transport within the Brewer-Dobson circulation (BDC). The deep branch of the BDC, comprising of rising tropical air reaching the mid and upper stratosphere, moving poleward and descending at high latitudes, is most relevant for ozone and climate in the polar regions (Lin and Fu, 2013; Butchart et al., 2014).

Figure 6 shows timeseries of the Arctic mean $\overline{w^*}$ (the vertical component of the residual mean meridional circulation in the Transformed Eulerian Mean framework, see Andrews et al., 1987) at 30 hPa in DJF (note positive values indicate upwelling). There is a mean increase in downwelling over the polar cap from ~1.8 mms$^{-1}$ to ~2 mms$^{-1}$ over the 1981-2080

period at a rate of ~0.015 (±0.007) $mms^{-1}$ $decade^{-1}$. Such an enhancement of the BDC is found in most CCMs (Butchart et al., 2010; Weber et al. 2011; Oberländer et al., 2013; Lin and Fu, 2013; Hardimann et al., 2014). A strengthening of the circulation will increase transport of ozone into the high latitudes, thereby contributing to the increase in springtime ozone over the century (Fig. 1(a)). In addition, it will drive adiabatic heating that would tend to offset the radiative cooling from

increasing $CO_2$. This in turn will impact on the Arctic temperature trends discussed in Sect. 3.2.2, in agreement with Butchart et al. (2010) and Langematz et al. (2014). These authors reported statistically not significant mid-winter (DJF and JF mean, respectively) temperature trends in the Arctic lower stratosphere, which they postulated to arise due to the compensation effect between the radiative cooling and dynamically driven warming.

In addition to the long-term trend in wintertime $\overline{w^*}$ over the Arctic, there is also large interannual and interdecadal variability

throughout the century. For example, we find a series of winters in the 2060s and 2070s in which the downwelling is anomalously weak. These coincide with years with an anomalously strong and cold polar vortex (not shown) and anomalously low Arctic column ozone in spring (Fig. 1(a)). Our model calculations therefore suggest that the dynamical conditions favouring low springtime ozone are expected to continue to occur in the future. There is also a period in the 2050s when most of the ensemble members show relatively enhanced downwelling over the Arctic (Fig. 6) and higher column

ozone amounts (Fig. 1(a)). Investigating the detailed drivers of this interdecadal variability is beyond the scope of this study, however, the results highlight the importance of both long-term trends and interannual and interdecadal variability for the future evolution of Arctic ozone.

Manney et al. (2011) and Langematz et al. (2014) have stressed that the occurrence of large ozone depletion episodes in the Arctic depends not only on the strength of the vortex during mid-winter, but also on its persistence into spring. A relatively

long-lived vortex extends the period in which temperatures fall below the PSC formation threshold, thereby allowing for continued halogen activation, substantial denitrification, as well as postponed resupply of ozone to the pole following the vortex break-up. It is therefore important to consider possible future changes in the strength as well as formation and break-up dates of the vortex.

Figure 7 shows linear trends in zonal mean zonal wind (u) at 61°N in (a) November/December and (b) March. In early

winter, the ensemble mean trend shows a strengthening of the winds in the lower and mid-stratosphere of ~0.1-0.25 $ms^{-1}$ $decade^{-1}$. Langematz et al. (2014) analysed the timing of the formation of the NH polar vortex and found a statistically significant trend towards earlier vortex formation. It is possible that the strengthening of the stratospheric zonal wind in autumn/early winter in our ensembles could be related to a similar effect. Importantly, there is at least one ensemble member that shows near-zero wind changes in the lower/mid stratosphere (100-30 hPa) and/or a weakening of the westerlies above,

highlighting the challenge of extracting robust trends in the presence of large dynamical variability. The ensemble spread is much larger in March (Fig. 7(b)) and, consequently, no statistically significant ensemble mean trends in the vortex strength can be found (see also Langematz et al.,2014).

We find that the large interannual dynamical variability that characterises the winter-time Arctic stratosphere is expected to persist in the future. In consequence, individual years characterised by dynamical conditions facilitating low column ozone

will continue to occur in the future. In the presence of declining stratospheric halogen levels, this dynamical variability will become increasingly important for determining interannual variability in Arctic spring total column ozone. However, as discussed in Sect. 3.2.1, halogen chemistry will also have a role to play. This role will reduce during the next several decades but will nevertheless remain important.

## 3.3 Case study of exceptionally low and average ozone events

The previous sections have examined the future behaviour of Arctic ozone in the UM-UMKCA ensemble, as well as its chemical and dynamical drivers. To further illustrate the potential impact of these processes in individual winters we now present a case study from the ensemble of an anomalously low total ozone event that occurs many decades into the future. Although the mean ozone column in the Arctic region increases steadily throughout the 21$^{st}$ century (Fig. 1(a)), individual winters continue to occur in which Arctic mean ozone is up to ~50-100 DU lower than the corresponding long-term mean. The green point in Figure 1(a) highlights the model year 2063 in one of the ensemble members. This particular year has an Arctic mean March ozone column of 383 DU, which is lower than the long-term minimum of the ensemble mean found in the late 1990s. We compare this to a model year with near average total ozone from the same period (2060, marked by orange point in Fig. 1(a)), whose March Arctic mean ozone column of ~489 DU is approximately 100 DU higher than in the model year 2063. These two case study model years are analysed below. Note that the years 2063/2060 are used here as a naming convention, rather than in reference to some specified years in the future.

Figure 8 shows timeseries of the daily Arctic mean total column ozone during the two case study winters of the model years 2063 (solid red) and 2060 (solid black). Column ozone increases throughout autumn and winter in both years due to transport of relatively ozone rich air from the tropics by the BDC (Strahan et al., 2013). Note that the Arctic mean encompasses not only the stratospheric polar vortex, but also some regions outside of it. For this reason, Fig. 8(a) also shows the vortex averaged column ozone for the two model years for comparison (dashed lines), defined here by the geographic region where daily mean Ertel's potential vorticity on the 850 K potential temperature surface is greater than or equal to $6\times10^{-4}$ m$^2$ s$^{-1}$ K kg$^{-1}$. These show that similar but somewhat smaller differences between the two years are found when the vortex-average column ozone is considered, compared with the Arctic mean.

The differences in column ozone between the two model years are smaller during late autumn and early winter than in subsequent months; this is in broad agreement with comparisons of similar years in observations (Strahan et al., 2013). Larger differences between the two years occur from late December onwards. In 2063, column ozone increases more slowly between mid-winter and spring, particularly in the Arctic mean. On 1 February, the Arctic mean column is 354 DU in 2063, which is 67 DU lower than in 2060. This is associated with an anomalously strong and cold polar vortex (see Fig. 8(b) and 9(b)), and relatively weak stratospheric wave driving in early winter (not shown). The strong polar vortex is associated with reduced downward transport of air inside the vortex. Consistently, the DJF mean 30hPa $\overline{w^*}$ is a mere -0.95 mms$^{-1}$, which is ~1.05 mms$^{-1}$ less negative than the 11-year running mean for that period (Fig. 6).

The differences in column ozone between the two model years are even larger in late winter and spring. In 2063, the polar vortex remains strong until early April (Fig. 9(b)), and total ozone reaches only ~390-395 DU by the end of March. The vortex average column ozone levels off and oscillates around ~360 DU until the beginning of the vortex break-up in mid-April (Fig. 9(b)). In contrast, in 2060 the polar vortex weakens substantially in mid-February and does not fully recover before the transition to summertime easterlies occurs (Fig. 9(a)). Consequently, the Arctic mean column ozone rises to ~490-500 DU by the end of March.

Figures 10(a) and (b) show time-altitude cross-sections of the differences in Arctic mean ozone mixing ratio, and ozone concentrations, respectively, between the model years 2063 and 2060. Small differences in ozone of up to ~0.5-1 ppm and ~$1 \times 10^{-12}$ molecules cm$^{-3}$ are already present in late autumn at ~25 km, which contribute to the differences in column amounts evident in Fig. 8(a). However, larger differences appear in the mid and upper stratosphere from the end of December onwards. This is particularly evident in the mixing ratios, with up to ~2.5-3 ppm less ozone between 30 and 40 km in 2063 compared to 2060. However, owing to the exponential decay in pressure with altitude the deficit is less pronounced in terms of absolute ozone amounts (up to ~$0.5 \times 10^{-12}$ molecules cm$^{-3}$). This anomaly is predominantly dynamically driven and is transported downward over the course of the winter, with differences in March maximising at 25-30 km. Ozone abundances are also reduced in the lower stratosphere in 2063 throughout winter and spring. Even though the differences in ozone mixing ratios at these levels are smaller compared with those in the mid and upper stratosphere, the higher ambient pressure results in significant changes in absolute ozone amounts. These therefore make a substantial contribution to the difference in total ozone columns in Fig. 8(a). In general, the deficits at all altitudes magnify from mid-winter to early spring, with ~$1-1.75 \times 10^{-12}$ molecules cm$^{-3}$ less ozone in the 10-30 km region in 2063 compared to 2060 (Fig. 10(b)). The difference in partial ozone column up to 25 km contributes a relative decrease of ~70 DU by the end of March. These are the altitudes where halogen catalysed ozone destruction plays its most significant role, as evidenced by elevated ClO concentrations in 2063 (Fig. 10(c)).

Figure 8(b) shows timeseries of the Arctic mean temperature at 22.7 km, in the region of NAT PSC formation, in 2060 (black) and 2063 (red). In 2060, the temperatures during early and mid-winter fluctuate around 210 K, but do not drop below ~203 K in the Arctic mean. In contrast, the Arctic mean temperature in 2063 oscillates near or below 195 K during most of January and February. The minimum local daily mean temperatures in 2063 reach 181 K (c.f. 191 K in 2060). The low temperatures allow the formation of PSCs (see red star in Fig. 5, Sect.3.2.3, for the potential $V_{PSC}$), which in turn permits heterogeneous chemical reactions that lead to the activation of chlorine from its reservoir species, HCl and ClONO$_2$, into reactive forms (ClO and Cl$_2$O$_2$, see Fig. S3, Supplementary Information). In early winter in 2063, ClO increases predominantly near the sunlit vortex edges, which could contribute to some ozone loss in the mid-latitudes (Pyle et al., 1994; Millard et al., 2002). From mid-January onwards, Arctic mean ClO levels in the lower stratosphere are ~125-150 ppt at 21.5 km (Fig. S3) and ~100-125 ppt at 19 km (not shown). This is up to ~75-125 ppt higher than in 2060 (Fig. 10(c)). While the Arctic mean values are relatively modest, local daily mean ClO levels in March reach up to ~300 ppt at 19 km in 2063 (c.f.

~80 ppt in 2060). The very low temperatures in the model year 2063 also lead to substantial denitrification such that the formation of PSCs, and halogen activation, from mid-winter is strongly reduced (not shown).

The dashed blue line in Fig. 8(a) shows vortex-average column ozone amounts in 2063 for a passive stratospheric ozone tracer, which when compared with the full (chemistry and transport) vortex-average ozone field enables a quantification of the relative roles of transport and (all) chemical processes in determining column ozone (see Tegtmeier et al. (2008) for an alternative approach). This passive tracer is initialised to the ozone field on 1 November and undergoes no chemical production or loss, but is advected by the circulation. The full ozone field and the passive tracer for 2063 track each other quite well, particularly in early and mid-winter. By the end of March both are relatively low and different from the ozone calculated in 2060 and it seems, therefore, that a major reason for the low column ozone in 2063 is dynamical in origin. There is also a difference between the full and passive ozone columns in 2063, which increases with time, with the cumulative effect of all chemical processes in the preceding five months averaged over the polar vortex contributing a 24 DU decrease to the total ozone column by the end of March in that year.  This shows that while dynamical processes are the dominant factor contributing to the low column ozone in 2063, chemical processes can not be neglected. We caution, however, against over-interpretation of this value as it is the result of a complex balance between different chemical loss and production cycles throughout the depth of the atmosphere as well as their interaction with transport throughout the winter.

Using our other diagnostic, i.e.  the cumulative (1. November to 30. March) halogen induced ozone loss in the lower atmosphere (up to 25 km), we find that the 6 main halogen cycles (see Sect. 2.3) result in 39 DU ozone loss in 2063 compared to 18 DU in 2060 (with very similar values of 44 DU and 23 DU, respectively, if we take vortex average; see also Table S2 and Fig. S4-S5, Supplementary Information, for other definitions of the polar vortex edge and stereographic plots of key quantities simulated on 1 March in the two model years). By this measure, the difference in estimated halogen losses between the two model years (~20 DU) is equivalent to about 20% of the full difference (~100 DU, see above) in the end of March Arctic mean total column ozone between the two years. Whilst the two diagnostics (i.e. chemical ozone loss derived from the passive ozone tracer and the halogen-induced ozone loss in the lower atmosphere) are not equivalent, both indicate that dynamical processes are likely to be a dominant driver of future interannual variability in Arctic ozone. However, as illustrated by the later case, loss of ozone through halogen chemistry remains a smaller but nonetheless important contributor to the modelled Arctic ozone variability even in the second half of the 21[st] century.

## 4. Conclusions

On its own, the projected decline in stratospheric $Cl_y$ over this century is expected to lead to the recovery of stratospheric ozone to pre-1980s values. However, a complex balance of chemical and dynamical processes will determine the precise evolution of Arctic ozone. For example, some studies have suggested that the Arctic lower stratosphere has become colder, with an increase in the incidence of temperatures low enough for the formation of PSCs (Rex et al., 2004, 2006). This could be related to the recent observation of particular years with exceptionally low Arctic springtime total column ozone (e.g.

Manney et al., 2011). If there is indeed a trend, a key question is whether it is a forced response to increased greenhouse gas concentrations. However, other studies (e.g. Rieder and Polvani, 2013) have argued that any observed changes in the Arctic stratosphere are consistent with natural variability. How the Arctic stratosphere and Arctic springtime ozone will evolve in the future therefore remains a key research question.

In this study, future trends in Arctic springtime total column ozone are assessed using a 7 member ensemble from the UM-UKCA CCMI REFC2 integrations. The Arctic mean March total column ozone increases throughout the 21$^{st}$ century in the simulations at a rate of ~11.5 DU decade$^{-1}$, and is projected to return to the 1980 level in the late 2030s. This is consistent with the long-term reduction in Cl$_y$ levels, GHG-induced stratospheric cooling as well as strengthened large scale meridional circulation, and is in overall agreement with previous modelling studies (Eyring at al., 2010; WMO, 2011, 2014).

Importantly, the integrations indicate that the high interannual variability that characterises the Arctic stratosphere today will persist in the future. Even beyond 2060 ozone can episodically drop ~50-100 DU below the corresponding long-term mean to near present-day values. These events coincide with an anomalously cold and strong polar vortex and reduced downwelling over the polar cap. These dynamical conditions are associated with reduced transport and mixing of ozone into the Arctic region and enhanced formation of PSCs which promotes activation of chlorine from its reservoir forms.

With regard to long-term trends, the ensemble mean shows a statistically significant cooling throughout most of the polar stratosphere in early winter, in agreement with the findings of Langematz et al. (2014). However, individual ensemble member can show near-zero temperature trends in the lower stratosphere, highlighting the need for ensemble studies for confident detection of trends in the polar lower stratosphere. During mid-winter, the inter-ensemble spread is larger and a significant cooling trend in the ensemble mean is only identifiable at pressures less than 30 hPa. In March, while all the

ensemble members indicate cooling of the mid and upper stratosphere, the inter-ensemble spread is large in the lowermost stratosphere at 100 hPa and no ensemble mean temperature trend can be found.

The results indicate a strengthening of the deep branch of the Brewer Dobson circulation in boreal winter, which is manifested by an increase in downwelling over the Arctic from December to February of ~0.015 (±0.007) mms$^{-1}$ decade$^{-1}$ (at 30 hPa); in agreement with previous modelling studies (Butchart et al., 2010; Weber et al., 2011; Oberländer et al., 2013; Lin

and Fu, 2013; Hardimann et al., 2014). Such an increase in the large-scale circulation will increase transport of ozone into the high latitudes, thereby contributing to the long-term increase in the spring Arctic ozone. In addition, it will also drive adiabatic warming in the Arctic stratosphere, thereby offsetting the radiatively-driven cooling from increasing CO$_2$. This compensation may contribute to the lack of statistically significant temperature trends in the lower stratosphere in mid-winter, in agreement with the results of Langematz et al. (2014). The simulations also show a small strengthening of the

zonal mean zonal winds in the extratropical lower and mid stratosphere in early winter of ~0.1-0.25 ms$^{-1}$ decade$^{-1}$, but no significant trends in March.

Consistent with the decline in Cl$_y$ over the century, the estimated halogen induced chemical ozone loss in the lower atmosphere decreases by around a factor of two between 1981-2000 and 2061-2080. This indicates a reduced role of halogen chemistry as a driver of springtime Arctic ozone and its variability. However, there are still individual winters later in the

century when dynamical conditions are favourable for the occurrences of elevated halogen losses. We conclude that despite the future reduction in levels of stratospheric halogens, chemical processes will play a smaller but non-negligible role in determining Arctic springtime column ozone throughout the century.

These points are exemplified by a case study of a low ozone year simulated in the 2060s. Even though the ensemble mean March Arctic $Cl_y$ in the lower stratosphere has dropped by ~50% with respect to the year 2000, the column ozone in this year is ~100 DU lower than the 11-year running mean and is close to the long-term minimum modelled near the turn of the century. This particular modelled winter is characterised by an anomalously strong and cold polar vortex that persists largely undisturbed until mid-April. Significantly suppressed transport reduces ozone supply throughout the winter. In addition, the anomalously low temperatures facilitate PSC formation and subsequent chlorine activation. The difference in the estimated halogen losses between this low ozone year and a year from the same period with near average springtime ozone is equivalent to ~20% of the difference in March total ozone column between the two years.

To conclude, our integrations suggest that the long-term recovery of Arctic ozone, driven by the future reduction in stratospheric $Cl_y$, GHG-induced cooling, and the increased strength of the large-scale stratospheric meridional circulation, means that the likelihood for individual years with springtime Arctic ozone depletion as severe as that observed in 2011 (e.g. Manney et al., 2011) will decrease in the future. However, the large interannual variability that characterises the Arctic polar vortex is expected to continue and this is likely to facilitate significant deviations of March ozone columns from long-term background values. Whilst our results suggest that dynamical processes will continue to play an important role for determining Arctic springtime ozone in the future, halogen chemistry will remain a smaller but non-negligible contributor for many decades to come.

**Acknowledgements**

The authors would like to thank James Keeble for constructive discussion and help with data handling. The authors thank Alex Archibald for helpful discussion and Paul Telford for his contribution to the model development as well as preparing and running the nudged UMUKCA CCMI REFC1 CheS+(SD) integration discussed in Sect. 2.1. We thank Russell Currie for transferring the data. We thank Katja Matthes for providing us with CMIP5 spectral solar irradiance. We would like to thank Greg Bodeker of Bodeker Scientific for providing the combined total column ozone database. We thank three anonymous referees for their constructive comments.

We thank NCAS Computational Model Support for help with setting up and porting the model. We acknowledge the ARCHER UK National Supercomputing Service. We acknowledge use of the MONSooN system, a collaborative facility supplied under the Joint Weather and Climate Research Programme, which is a strategic partnership between the UK Met Office and the NERC.

ACM, JAP and NLA were supported by the National Centre for Atmospheric Science a NERC funded research centre. We acknowledge funding from the ERC for the ACCI project grant number 267760, including a PhD studentship for EMB. ACM acknowledges support from an AXA Postdoctoral Fellowship and NERC grant NE/M018199/1.

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

Figure 1. Timeseries of monthly mean polar averaged (65-90°) total column ozone [DU] from 1960-2100 for (a) the Arctic in March, and (b) the Antarctic in October. (c) As in (a) for the March total inorganic chlorine loading ($Cl_y$) at 20 km [ppb]. The red lines in (a) and (b) denote observations from the Bodeker total ozone column dataset (Bodeker et al. (2005); Müller et al. (2008)). Thick black and blue lines denote the ensemble mean and its 11-year running average, respectively. Orange and green points in (a) denote the two case study years of 2060 and 2063, respectively, described in Sect. 3.3.

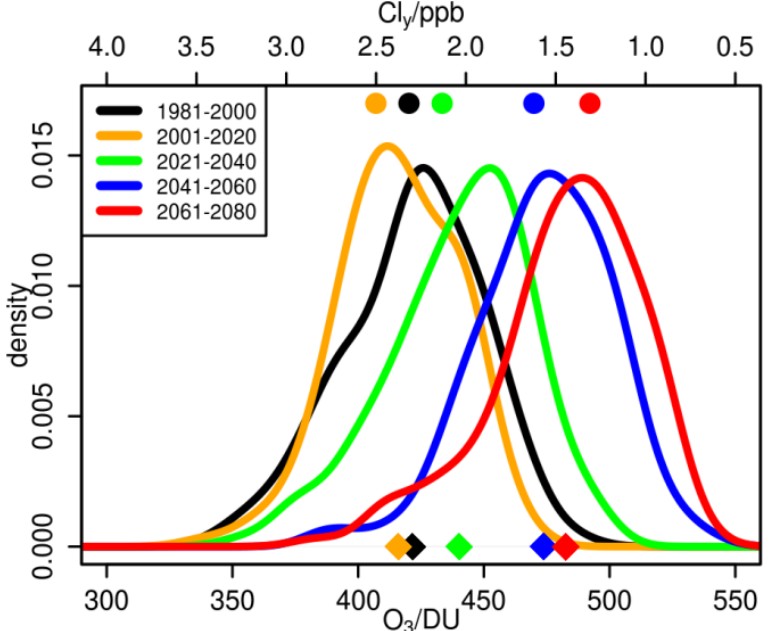

Figure 2. Probability density functions (PDFs) for 20-year intervals taken from the period 1981 to 2080 for Arctic mean total ozone column [DU] in March. Each PDF contains data from 7 ensemble members. Coloured diamonds indicate the 20-year means of the total ozone columns and coloured points the corresponding means of Arctic mean $Cl_y$ [ppb] at 20 km.

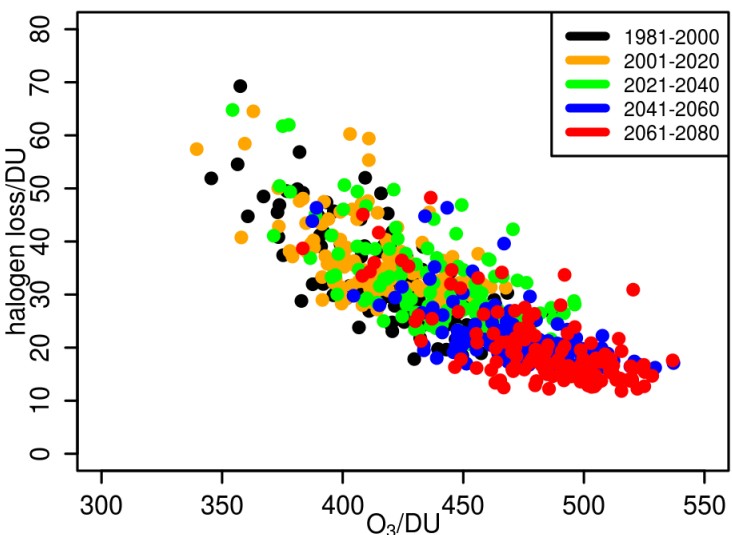

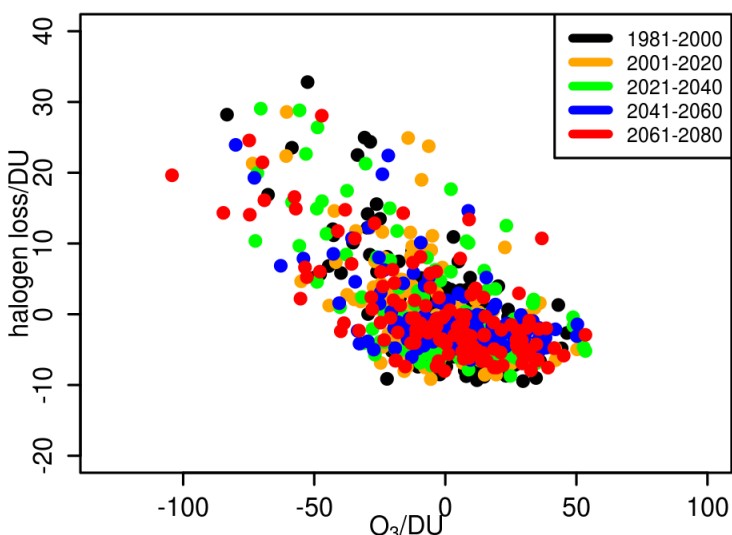

Figure 3. (a) Scatterplot of Arctic mean March total column ozone [DU] versus cumulative (1 November to 30 March) halogen induced total ozone loss [DU] (see Sect. 2.3) summed from the surface to 25 km. Each point shows a single winter and the colours denote the same 20 year intervals as in Fig. 2. (b) As in (a) but for deviations of column ozone and halogen losses from their corresponding 11-year running ensemble mean.

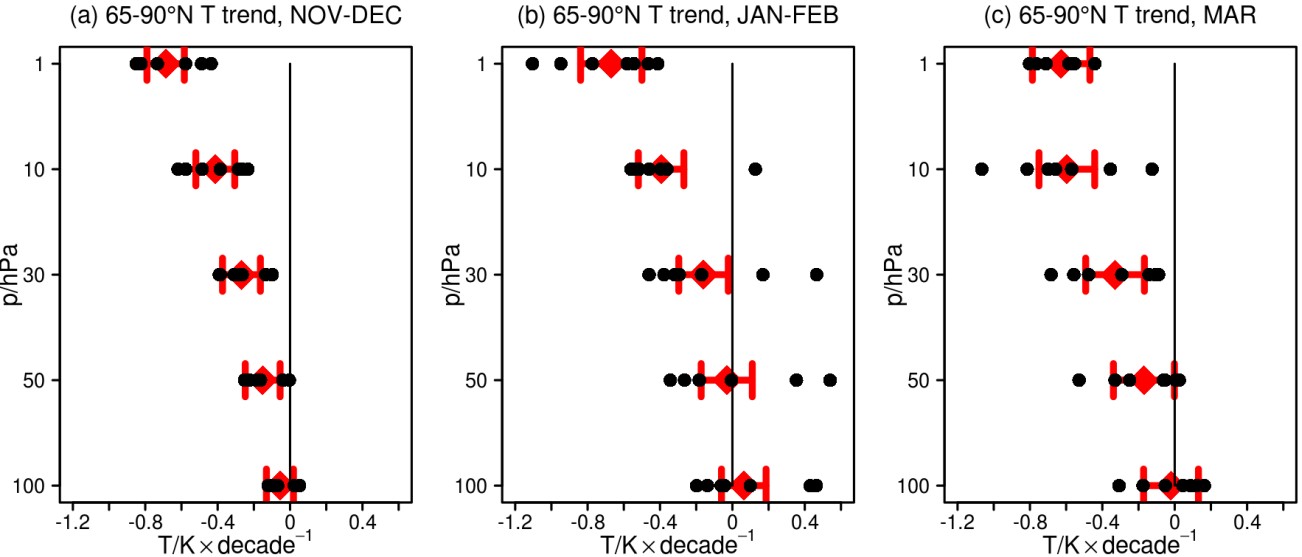

Figure 4. Linear trends in Arctic mean temperature [K decade$^{-1}$] at selected stratospheric pressure levels in (a) November/December, (b) January/February and (c) March calculated for each ensemble member individually (black points) for the period 1981-2080. Red diamonds and whiskers show the corresponding trends (±2 standard errors) calculated for the ensemble mean.

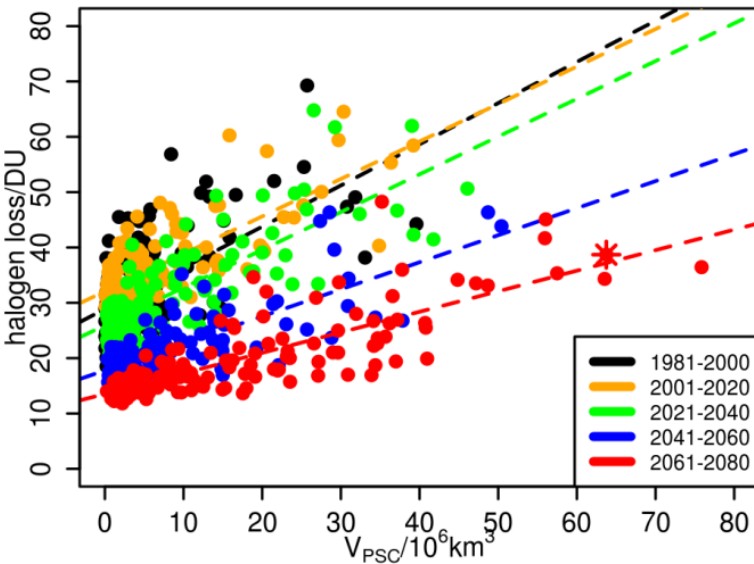

Figure 5. Scatterplot for November to March mean potential $V_{PSC}$ [$\times 10^6$ km$^3$] against halogen induced ozone loss [DU] over 1-25 km integrated over the same months. Colours denote the same 20 year intervals as in Fig. 2. Dashed lines indicate linear fits for the individual 20 year periods. The model year 2063 analysed in Sect. 3.3 is highlighted with a red star.

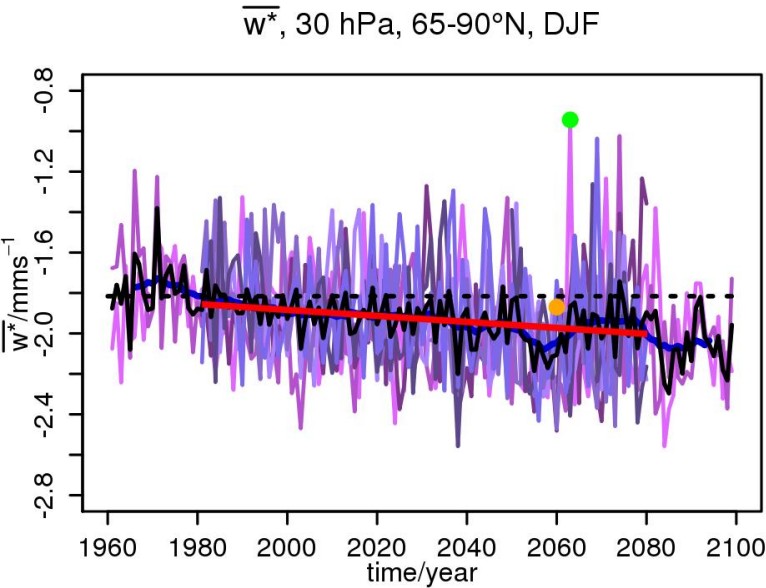

Figure 6. Timeseries of DJF mean residual vertical velocity, $\overline{w^*}$ [mms$^{-1}$], over 65-90°N at 30 hPa from 1960-2100. Thick black and blue lines denote the ensemble mean and its 11-year running average, respectively. Thick red line shows the linear trend calculated for the ensemble mean over the 1981-2080 period. Orange and green points denote the two case study model years 2060 and 2063, respectively, discussed in Sect. 3.3.

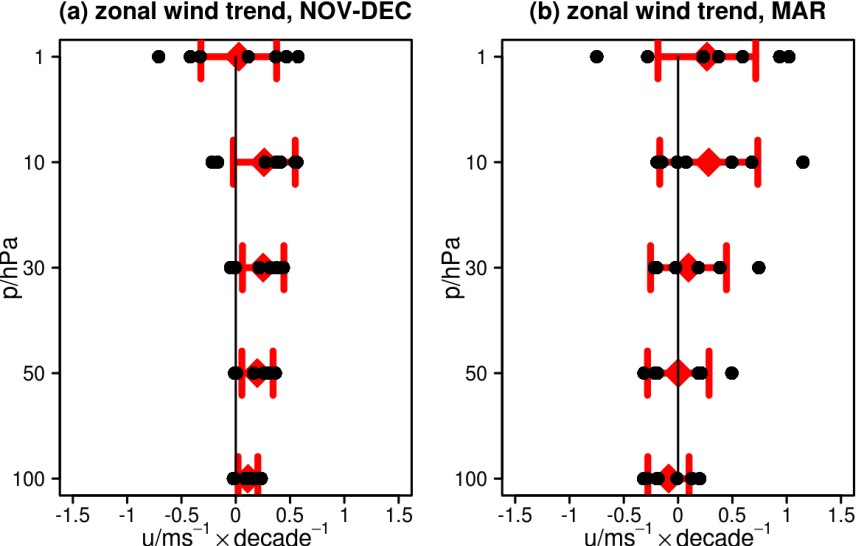

Figure 7. Linear trends in zonal mean zonal wind [ms$^{-1}$ decade$^{-1}$] at 61°N at selected stratospheric pressure levels in (a) November/December and (b) March calculated for each ensemble member individually (black points) for the period 1981-2080. Red diamonds and whiskers show the corresponding trends (±2 standard errors) calculated for the ensemble mean.

## (a) total ozone column

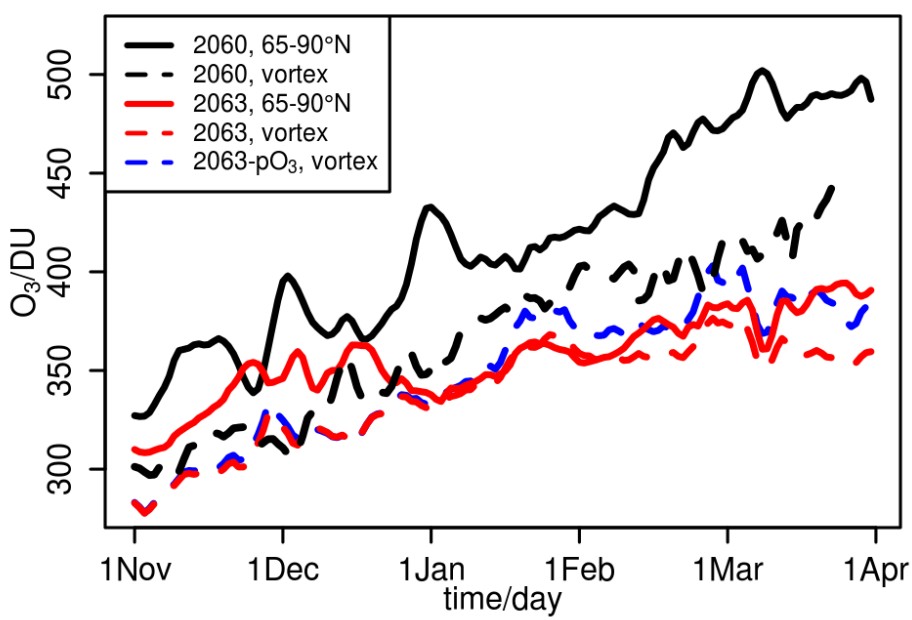

## (b) temperature at ~22.7 km

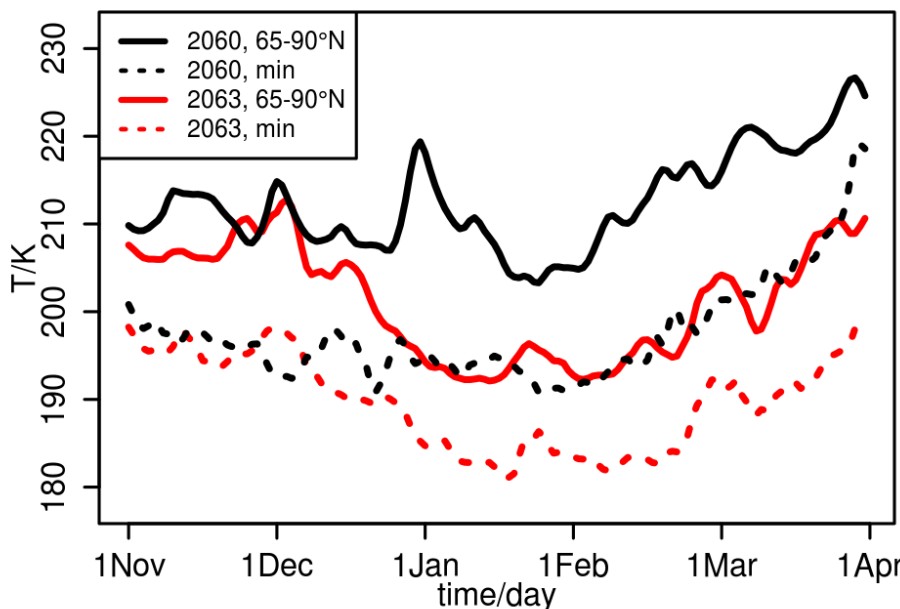

Figure 8. Timeseries of Northern Hemisphere (a) daily total column ozone [DU] during the winters 2060 (black) and 2063 (red), and (b) temperature [K] at 22.7 km. Solid lines show the Arctic mean (65-90°N), dashed lines in (a) show polar vortex averages (see text for details) and dotted lines in (b) show minimum daily mean temperatures found anywhere poleward of 65°N. The blue dashed line in (a) shows the evolution of the vortex-averaged passive ozone tracer in 2063.

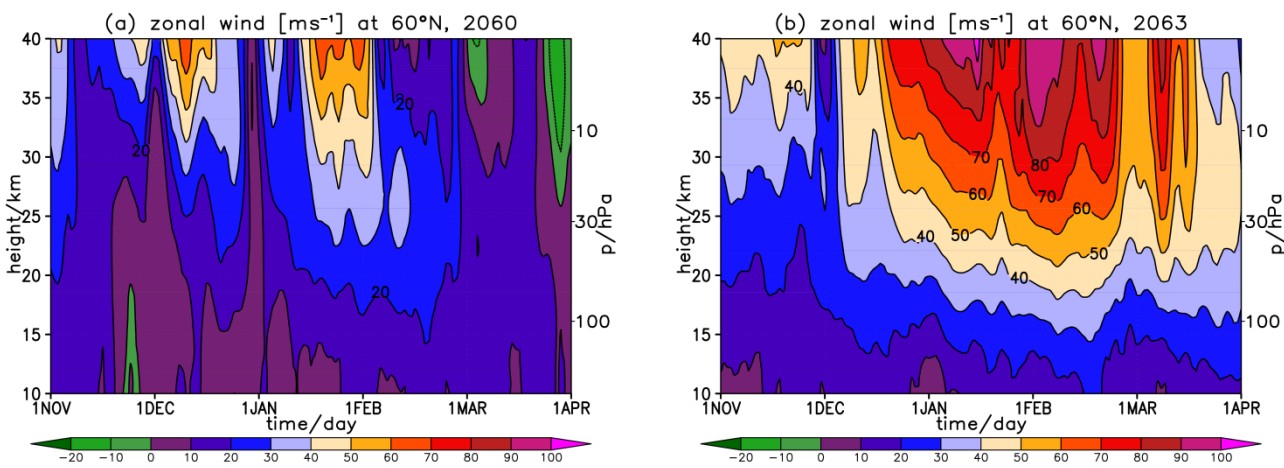

Figure 9. Time-altitude cross-sections of the daily Arctic (60°N) zonal mean zonal wind [ms$^{-1}$] in (a) 2060 and (b) 2063.

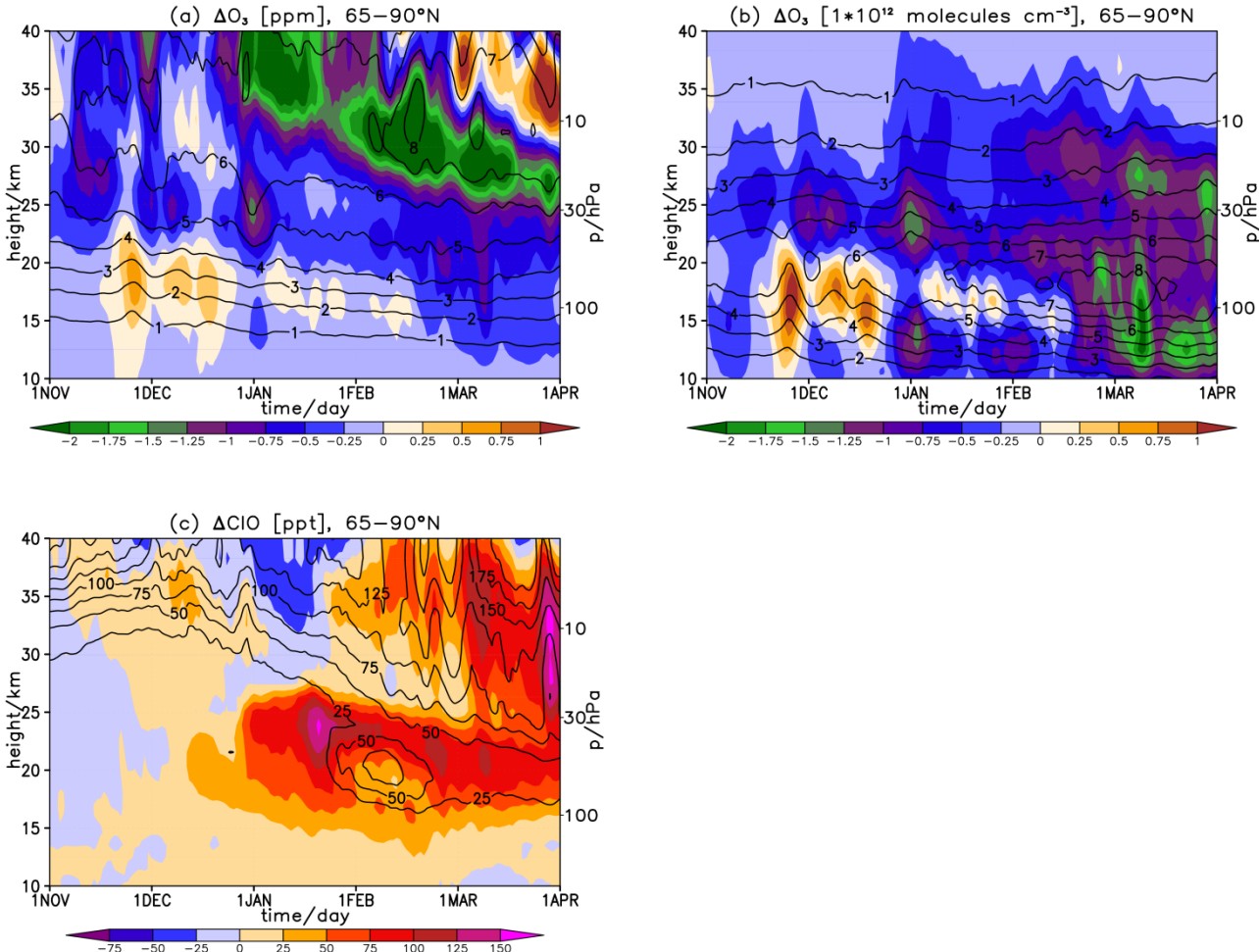

Figure 10. Colours: Time-altitude cross-sections of the Arctic mean differences between the years 2063 and 2060 in (a) ozone mixing ratios [ppm], (b) ozone number density [$10^{12}$ molecules cm$^{-3}$] and (c) ClO mixing ratio [ppt]. The solid contours show the year 2060 for reference.

