# Peer review of "Future Arctic ozone recovery: the importance of chemistry and dynamics"

_Atmospheric Chemistry and Physics, 2015_

## Referee Comment (RC1) · Anonymous Referee #1 · 7 Feb 2016

*Review on the manuscript „Future Arctic ozone recovery: the importance of chemistry and dynamics" by E. M. Bednarz, A. C. Maycock, N. L. Abraham, P. Braesicke, O. Dessens, and J. A. Pyle for Publication in ACP*

This paper deals with the recovery of Arctic ozone in a future climate under increasing greenhouse gas concentrations and declining inorganic chlorine. In contrast to earlier studies the authors use an ensemble mean of seven transient simulations to capture the interannual variability in Arctic ozone. The special focus is on the possibility of individual years with strong ozone depletion even after 2060 when halogen loading has become relatively low.

I find the paper appropriate for publication in ACP after my minor suggestions have been considered and my questions have been clarified.

**General comments:**
- The authors need to be very careful with the references in the text pointing to the single figures. There are some mistakes which can confuse the reader.
- It need to be clearly stated that the years '2060' and '2063' are model years coming out of a not nudged simulation and are therefore relatively arbitrary. In some parts it sounds as if we get a really strong ozone loss in the future year 2063 and a really weak one in 2060. This needs to be clarified.
- The authors have included a lot of citations in their manuscript and compare their results with many of these studies. In some cases they need to be more specified. In my opinion some studies do not exactly show what is stated here.
- There are a lot of typos in the references. A cross check should be done before final publication.

**Specific comments:**
- Page 4, line 27: How are orographic and non-orographic gravity waves parameterized? Please provide a reference.
- Page 5, lines 15 - 17: I don't really understand why these six year bins are excluded. The supplement only shows which years are affected. Please provide some more information on this, here or in the supplement.
- Page 6, line 11: 'In all six reactions, a net loss of 2 odd oxygen molecules occurs per cycle.' → Do you mean 2 molecules ozone? But in cycle 3 there is only one. Please clarify this.
- Page 6, line 24: 'averaged from 65°N to the pole.' → Why do you use exactly 65°N - 90°N? Have you also tested other latitudes, for example 60°N - 90°N and does this change the results?
- Page 7, line 6: see comment above
- Page 10, lines 17 - 19: '...in agreement with Langematz et al. (2014).' → This statement should be specified. Do you compare with Figure 2a from Langematz et al. (2014)? From this figure I see a significant trend at 100 hPa, which is not the same as in your study. Moreover, you have to note that the time ranges are not identical.
- Page 11, line 26: 'This is in broad agreement with the findings in Langematz et al. (2014).' → Where do you get this from? The focus in their study is on the vortex duration and not on the zonal wind trend. You need to be more specific with your comment.

- Page 12, line 3: Maybe you can call Section 3.3 'Case studies of exceptionally low and high ozone events' as you show results from both - low and high - and not only from low ozone events. This should be changed also in the Introduction (page 4, lines 5 - 7).
- Page 12, Section 3.3.: As you use free running, and no nudged model simulations, you won't expect that your 'model' years resemble 'real' years. Please make sure that the 'years' 2060 and 2063 are 'model' years. Do you really need this numbers? Maybe you can skip them and refer to low and high ozone events.
- Be very careful with the references on the figures:
  - Page 12, lines 25 - 26: ...(see Fig. 7(b) and 8 (b)).
  - Page 12, line 30: (Fig. 8(b)) and not 8(a)!!!
  - Page 12, line 31: (Fig. 8(b)) and not 8(a)!!!
  - Page 13, line 1: (Fig. 8(b)) and not 8(a)!!!
  - Page 13, line 26: (Fig. 9(c)) and not 9(b)!!!
- Page 15, line 31: '...account for ~20%...' → This is a very crude estimate. Please be more specific.

**Technical corrections:**
- Page 2, lines 9 - 10: reference for 'Montreal Protocol on Substances that Deplete the Ozone Layer'
- Page 3, line 19: ... volume of *PSCs ($V_{PSC}$)* → the abbreviation 'PSCs' has been introduced in line 9
- Page 4, line 1: *Stratosphere -troposphere* Processes *And* their Role in Climate (SPARC)
- Page 4, line 2: Please provide a reference for CCMI.
- Page 4, line 16: ... the recent SPARC Report on the Lifetimes of ... (SPARC, 2013;...) → Be careful that this is in line with the citation on page 21, line 14f.
- Page 4, line 24: The dot at the end of the sentence is missing.
- Page 5, line 13: You may introduce an abbreviation for 'sea-ice concentrations' here and use it on page 8, lines 18 and 27.
- Page 5, line 17: ...long periods *are* excluded...
- Page 5, line 23: ... and a more minor $ClO + O(^3P)$ cycle... → You should include (Cycle 3, *reference*) as before.
- Page 11, lines 3 - 4: The references should be sorted by year.
- Page 11, lines 29 - 30: ...(see also Langematz et al., 2014).
- Page 12, line 12: ... higher than in *model year* 2063.
- Page 12, line 15: use the abbreviation 'BDC', as introduced before
- Page 13, line 17: ... ClO concentrations in 2063 *compared to 2060* (Fig. 9(c)). → The figure shows a difference and not the concentrations in 2063.
- Page 15, line 26: ...'exemplified by a case study in 2063.' → Either you include 'model year' here, or you skip the year. In the Conclusions I would prefer to skip the years and use 'low and high ozone events instead.
- Page 15, line 32: '...in year 2063 and a year from the same period ... ' → Better: '...between this year and a year from the same period ...'
- Page 17, line 4: ... Steil, B.; *and* Tian, W....

- Page 17, line 5: The dot is missing at the end of the reference.
- Page 17, line 19: Drdla, K., and *Müller*, R.:...
- Page 19, line 10: *...Oberländer*, S., ...
- Page 21, line 30: Tilmes, S., *Müller*, R., ...
- Page 22, line 5: ... and *Müller*, R.: ...
- Page 24, line 4: ... 11-year running average*, respectively*.
- Page 24, line 5:... 2060 and 2063*, respectively,* described in Sect. 3.3.
- Page 27, line 2: ... 11-year running average*, respectively*.
- Page 27, line 6: 'As in Figure 4, ...' → I would prefer an independent figure caption for Figure 6, as the only agreements with Figure 4 are the pressure levels and the meaning of the points and bars.

---

## Referee Comment (RC2) · Anonymous Referee #2 · 18 Feb 2016

**Review of "Future Arctic ..."**

BY BEDNARTZ ET AL.

**General**

The paper addresses an important scientific question, namely projections of future ozone levels in the Arctic. Overall the paper is well written and the discussions and arguments are clear (see below for some exceptions). The study uses a well established and well described model (the UM-UKCA model). The results on the timing of future Arctic ozone recovery are very relevant for the readership of ACP and also for future scientific ozone assessments. A particular strength of the paper is the use of ensembles and the analysis of relevant chemical and dynamical processes with a focus on the case of the simulation for winter 2063.

One major question to the models projecting the future is how well they simulate present day chemical Arctic ozone loss. Of course, this is a prerequisite for the assessment of future recovery. And it is not obvious that state-of-the-art models do a good job in all respects at present day chemical Arctic ozone loss. For example, Brakebusch et al. (2013) find a systematic high bias in ozone in the model of 18% in the lowermost stratosphere in March. They attribute most of this ozone bias to too little heterogeneous processing of halogens late in the winter and suggest that the model underpredicts $ClONO_2$ early in the winter and has too little activated chlorine. How is the UM-UKCA model doing in this respect? How well does the model simulate denitrification (which is important for Arctic ozone loss)? Further, as far as I understand the UM-UKCA model uses an equilibrium NAT scheme, where NAT is formed at the NAT equilibrium temperature, which likely overestimates the onset of heterogeneous reactivity in the model. Are there earlier studies, where these points have been addressed?

In the model heterogeneous reactivity is driven by NAT and ice particles, i.e., what regards the Arctic largely by NAT. This is likely not realistic, as there are extensive observations of liquid particles in the polar regions (e.g., Pitts et al., 2013). Nonetheless, Keeble et al. (2014) obtain a reasonable simulation of the Antarctic ozone hole assuming heterogeneous reactions on NAT and ice using the model employed here. Similarly, Grooß et al. (2011) used a set-up with a NAT dominated heterogeneous chemistry (likely not realistic) but were able to reproduce the observed extremely low ozone values in the Antarctic. Possibly, it is not necessary to get every detail of PSC formation right to obtain a reasonable representation of chlorine activation and ozone loss in the model (see also Kirner et al., 2015; Solomon et al., 2015, and references therein). But I suggest that the issue of heterogeneous reactivity is discussed in more detail in the paper (see also detailed comments below).

Moreover, an important theme of the paper is halogen induced ozone loss due

to heterogeneous reactions and chlorine activation (and the relative role of dynamics). As sufficiently cold conditions develop almost exclusively in the polar vortex, halogen induced ozone loss is only expected to occur in the vortex. However the analysis in the paper is mostly based on geographical latitude thereby neglecting the distinction between inside and outside of the vortex (in Fig. 7a however, a vortex average is presented, see also comments below). For example, how different would Figure 1 look, if equivalent latitude would be used rather than geographic latitude? Further, in the discussions on the ozone anomaly simulated for the year 2063, the distinction between vortex processes and out of vortex processes is not always brought across clearly (see detailed comments below). Form my reading of the discussion in the paper (top of page 14), in the model, a significant fraction of the 2063 anomaly is driven by chemistry *outside* of the vortex – is this correct? I suggest improving the discussion and carefully quantify the contributions of chemistry and dynamics inside and outside of the polar vortex to the simulated ozone anomaly in 2063.

In summary, I think with respect to several issues raised in this review, the paper needs to be revised and improved. Nonetheless, I believe that this is a potentially very good paper, which could make an important contribution to improved projections of future Arctic ozone levels and in particular regarding the various processes impacting polar ozone. The paper will also be very relevant to the upcoming new WMO ozone assessment.

**Detailed comments**

- p 1, l 15: This statement is confusing: to me it implies that present day spring Arctic ozone is 50-100 DU below the values expected after recovery in 2060. Is this what you want to say here? Is this true for your model simulations presented here?

- p 1, l 20: why does an increase in downwelling lead to less consistency?

- p 2, l 2: The use of CFCs did not lead to the 'suggestion...'

- p. 2, l 4: One should distinguish the issue raised by Molina and Rowland (1974) (upper stratospheric ozone, globally) from the ozone hole issue pointed out by Farman et al. (1985).

- p 2., l9: a citation from 1997 does not really allow to say 'soon' with reference to Farman et al. (1985).

- p 3, l 1: the impact is also on ecosystems not only on human populations.

- p 3, l 14: change 'sulphate' to 'cold sulphate'

- p. 3, l 25: 'controlling' is perhaps to strong

- p 4, l 30: it might be worth pointing out that the mean age of the UM-UKCA model is in relatively good agreement with the observations in the high latitudes of the Northern hemisphere (according to Chipperfield et al., 2014), which is the most important region for this study.

- I am assuming that not only this reaction is not taken into account on liquid aerosol but all five reactions listed in on page 13707 of Keeble et al. (2014). While this is likely not realistic (there are extensive observations of liquid particles in the polar regions, e.g., Pitts et al., 2013) is should not affect the quality of the ozone loss simulations too much. Assuming that the details of heterogeneous reactivity are not essential for a good representation of polar ozone loss; see also discussion below.

- p. 5, l 10-12: I do not think it is correct to say that Keeble et al. (2014) showed that ozone depletion can be attributed to heterogeneous reactions on NAT and ice. They obtain a reasonable simulation of the Antarctic ozone hole making this assumption. In the real world, for a long time, the heterogeneous reactivity will be dominated by ice particles. On the other hand, neglecting ice particles (and indeed NAT) does not result in a substantial change of the simulated ozone loss (Kirner et al., 2015; Solomon et al., 2015). So I think the wording should be more careful here.

- p 5, l 29: chemical formulas should not be in italics

- Section 2.3: I would suggest some more discussion of the relevance of the cycles discussed here. Could you roughly quantify what is meant with "lesser importance". It could be close to negligible for some of the cycles I think. On the other hand close to the tropopause in the non activated region natural (e.g. $HO_x$ driven cycles might be important for ozone loss.

- Figure 1: how different would this figure look, if equivalent latitude would be used rather than geographic latitude. Would this not be the better choice? What is the reason for preferring geographic latitude vs. equivalent latitude?

- p 7, l 15: change 'atmospheric' to 'stratospheric' – $Cl_y$ is not defined in the troposphere

- p 7, l. 20: I do not agree. The paper by Haigh and Pyle (1982) does not discuss the relevant ozone loss cycles in the polar regions, in particularly the ClO-dimer cycle, which does not slow down with decreasing temperature. I think you are discussing polar ozone loss in the lower stratosphere here.

- p 7, l 31: If I understand correctly, this value is computed by determining the minimum ozone value poleward of 65°N each day and then computing the mean value over a month. Have you ensured that all these values are within the polar vortex? Or could some of these values stem from (dynamically caused) so-called mini-holes?

- p 8, l 5: What is the implication of this statement? This sentence could be interpreted as stating that under present day conditions routinely strongly depleted ozone values are found. Please clarify.

- P. 8, l 25-28: Difference to the results of Langematz et al. (2014); if the reason is 'differences in the representation', do you mean chemical or dynamical effects? If you agree with me that the difference is very likely not due to chemistry, you could state this point more clearly.

- p 9, l 6: This is a bit misleading – are there more, even less important halogen cycles? I suggest stating which of the six cycles are dominant, which play a minor role and which are negligible.

- p. 9, l 21: do you really mean 'halogen losses' here?

- p 9, l 26: this formulation is a bit awkward; I think you never applied the 11-year running mean rather than removing it.

- p 10, l 3: not only the amount of PSCs also the length of the cold period. This is also important (Manney et al., 2011). Further below you also make this point.

- p 10, l 18: change 'insignificant' to 'not significant'

- p 10, l. 19: "Similar is true" – reformulate

- p 10, l 27-29: Actually, the ozone levels in Antarctica are also strongly influenced by the BD-circulation; it is just that the dynamical variability is lower – correct?

- p 11, line 1: provide a citation and/or explanation for $\bar{w}^\star$

- p 11, l 15: 'relatively' to what?

- p. 11, l 21-22: A central issue here is also the continued presence of PSCs and thus the continued activation.

- p 11, l 32: will continue to occur in the future ...

- p 12, l 10: "long-term minimum of ensemble mean" – this is not quite clear? Which period is exactly considered? And what is meant is the lowest value for March mean ozone in the ensemble?

- p 12, l 16: This effect could be reduced by considering equivalent latitude.

- p 12, l 17-19: Why did you choose 850 K to define the vortex? This is above the altitude where most halogen induced ozone loss occurs. Also how is the PV value defined (citation?)?

- p 12, l 20: I suggest showing the vortex average data.

- p 13, l 9: this is an important point that should also be brought across clearly in the abstract.

- p 13, l 18: as stated before, heterogeneous reactivity in general should be more important than NAT formation in particular. Also, is there any formation of ice particles in the model for the year 2063?

- p 14, l 2-4: it is interesting to note that only part of the effect of the anomaly has its origin in processes in the polar vortex. Doesn't this mean that in the model only part of the chemistry driven effect is caused by halogen chemistry? Again I suggest to bring this message more clearly across the the abstract.

- p 14, l 9: here you state that the halogen effect is 40 DU but above you state that the polar vortex effect is only 25 DU. Does this mean that in the model, a significant fraction of the 2063 anomaly is driven by halogen chemistry *outside* of the vortex? Is there chlorine activation outside of the vortex in the model? This discussion at this stage and the attribution of ozone loss to processes needs to be improved.

- p 14, l 21: citation for 'other studies'

- p 15, l 31: it is not clear to me where the number of 20% is coming from. on p. 14, you report that the halogen induced loss in 2063 is twice that of 2060, which might be the first order information of interest here. However, to me the question is still open of how much the difference between 2060 and 2063 is a polar vortex effect and in how far is is influenced substantially by out of vortex processes.

**References**

Brakebusch, M., Randall, C. E., Kinnison, D. E., Tilmes, S., Santee, M. L., and Manney, G. L.: Evaluation of Whole Atmosphere Community Climate Model simulations of ozone during Arctic winter 2004-2005, J. Geophys. Res., 118, 2673–2688, doi:10.1002/jgrd.50226, 2013.

Chipperfield, M. P., Liang, Q., Strahan, S. E., Morgenstern, O., Dhomse, S. S., Abraham, N. L., Archibald, A. T., Bekki, S., Braesicke, P., Di Genova, G., Fleming, E. L., Hardiman, S. C., Iachetti, D., Jackman, C. H., Kinnison, D. E., Marchand, M., Pitari, G., Pyle, J. A., Rozanov, E., Stenke, A., and Tummon, F.: Multimodel estimates of atmospheric lifetimes of long-lived ozone-depleting substances: Present and future, J. Geophys. Res., 119, 2555–2573, doi:10.1002/2013JD021097, URL http://dx.doi.org/10.1002/2013JD021097, 2014.

Farman, J. C., Gardiner, B. G., and Shanklin, J. D.: Large losses of total ozone in Antarctica reveal seasonal $ClO_x/NO_x$ interaction, Nature, 315, 207–210, 1985.

Grooß, J.-U., Brautzsch, K., Pommrich, R., Solomon, S., and Müller, R.: Stratospheric ozone chemistry in the Antarctic: What controls the lowest values that can be reached and their recovery?, Atmos. Chem. Phys., 11, 12 217–12 226, 2011.

Keeble, J., Braesicke, P., Abraham, N. L., Roscoe, H. K., and Pyle, J. A.: The impact of polar stratospheric ozone loss on Southern Hemisphere stratospheric circulation and climate, Atmos. Chem. Phys., 14, 13 705–13 717, doi:10.5194/acp-14-13705-2014, URL http://www.atmos-chem-phys.net/14/13705/2014/, 2014.

Kirner, O., Müller, R., Ruhnke, R., and Fischer, H.: Contribution of liquid, NAT and ice particles to chlorine activation and ozone depletion in Antarctic winter and spring, Atmos. Chem. Phys., 15, 2019–2030, doi:10.5194/acp-15-2019-2015, 2015.

Langematz, U., Meul, S., Grunow, K., Romanowsky, E., Oberländer, S., Abalichin, J., and Kubin, A.: Future Arctic temperature and ozone: The role of stratospheric composition changes, J. Geophys. Res., 119, 2092–2112, doi:10.1002/2013JD021100, URL http://dx.doi.org/10.1002/2013JD021100, 2013JD021100, 2014.

Manney, G. L., Santee, M. L., Rex, M., Livesey, N. J., Pitts, M. C., Veefkind, P., Nash, E. R., Wohltmann, I., Lehmann, R., Froidevaux, L., Poole, L. R., Schoeberl, M. R., Haffner, D. P., Davies, J., Dorokhov, V., Gernandt, H., Johnson, B., Kivi, R., Kyrö, E., Larsen, N., Levelt, P. F., Makshtas, A., McElroy, C. T., Nakajima, H., Parrondo, M. C., Tarasick, D. W., von der Gathen, P., Walker, K. A., and Zinoviev, N. S.: Unprecedented Arctic ozone loss in 2011, Nature, 478, 469–475, doi:10.1038/nature10556, 2011.

Molina, M. J. and Rowland, F. S.: Stratospheric sink for chlorofluoromethanes: chlorine atom catalysed destruction of ozone, Nature, 249, 810–812, 1974.

Pitts, M. C., Poole, L. R., Lambert, A., and Thomason, L. W.: An assessment of CALIOP polar stratospheric cloud composition classification, Atmos. Chem. Phys., 13, 2975–2988, doi:10.5194/acp-13-2975-2013, URL http://www.atmos-chem-phys.net/13/2975/2013/, 2013.

Solomon, S., Kinnison, D., Bandoro, J., and Garcia, R.: Simulation of polar ozone depletion: An update, J. Geophys. Res., 120, 7958–7974, doi:10.1002/2015JD023365, 2015.

---

## Referee Comment (RC3) · Anonymous Referee #3 · 23 Feb 2016

This article is based on an interesting study of the projection of future Arctic ozone using ensemble simulation from the UM-UKCA chemistry climate model. While other studies have been performed on this subject (e.g. WMO, 2011; Langematz et al., 2014), the originality of the study lies in the use of ensemble simulation, which allows the authors to estimate the intrinsic variability of the stratosphere, together with the impact of ozone depleting substances decrease and climate change on Arctic ozone. The paper is well written and informative for the projection of future Arctic ozone and I recommend publication in ACP, provided that important comments for improvement are taken into account.

Main comments

• The main focus of the study is on the respective contribution of chemistry and

dynamics on future Arctic ozone. In that respect diagnostics have been set up in order to evaluate the importance of chlorine chemistry in future ozone loss and the authors argue that halogen chemistry can still play a substantial role after mid-century. Since this result is rather intriguing, it deserves more attention in the article. A whole section is dedicated to the case study of winter 2063 but it is somewhat descriptive and does not demonstrate fully that the chemical loss is linked to halogen chemistry. For example, is the observed loss coherent with the known relationship between Cly levels, chlorine activation and PSC volume (e.g. Rex et al., 2004)? What is the role of nitrogen chemistry that can sometimes be important in the Arctic mid-stratosphere as shown in Kuttipurath et al., 2010? A quantification of PSC volume and a figure similar to Figure 2 but showing observations in order to demonstrate the skills of the model to simulate halogen chemistry would be useful.

• Since the Arctic ozone loss is computed over the 65-90°N latitude range, it encompasses some loss from non-vortex air. This issue is acknowledged by the authors but would need some quantification.

• From Figure 1, it seems that the interannual variability of Arctic ozone from the ensemble simulation is larger than the natural variability as seen from the observations. Can the authors comment on that and provide some statistics on this issue? In addition a more substantial description in section 2 of the skills of the UM-UKCA model in terms of polar ozone simulation is needed: e.g is there a cold bias of the polar stratosphere? How the strength and duration of the Northern vortex compare with observations, . . .?

• Temperature trends: No mention is made of the evolution of the occurrence of sudden warmings in the ensemble simulation. It is thus difficult to distinguish radiatively induced with dynamically induced temperature trends. This issue should be addressed.

Minor comments

P5 l24: it is not clear how the 11-year solar cycle is simulated over the 21st century.

P7 l21: the products of the reaction are wrong: it should be Cl + O2.

P6 l4: This sentence is not so clear. The computational efficiency relates to the use of the diagnostics for evaluating halogen induced ozone loss.

P9 l21: what is the contribution of slowing of gaz phase ozone loss cycles compared to changes in stratospheric transport in the earlier recovery of Arctic ozone?

P11 l4-13: In line with my major comments, the causes of the drops of Arctic ozone in late century, and the comparison with Langematz et al. (2014) study should be better substantiated.

P12 l10-18: a chemical loss of 40 DU is similar to the current Arctic ozone losses, while chlorine levels in 2061-2080 will be lower by more than a factor of 2. What PSC volume is necessary for such extreme loss?

P16 l13: what is the justification for the PV value to define the vortex?

Figure 5: Case study years (2060 and 2063) should be highlighted in the figure.

—————————————————————

---

## Author Comment (AC1) · 29 Jun 2016

**AUTHORS' RESPONSE TO THE REFEREE 1 COMMENTS**

We thank Referee 1 for helpful comments regarding improving our manuscript. Below are point by point replies to the particular issues raised.

*This paper deals with the recovery of Arctic ozone in a future climate under increasing greenhouse gas concentrations and declining inorganic chlorine. In contrast to earlier studies the authors use an ensemble mean of seven transient simulations to capture the interannual variability in Arctic ozone. The special focus is on the possibility of individual years with strong ozone depletion even after 2060 when halogen loading has become relatively low. I find the paper appropriate for publication in ACP after my minor suggestions have been considered and my questions have been clarified.*

*General comments:*
*☐ The authors need to be very careful with the references in the text pointing to the single figures. There are some mistakes which can confuse the reader.*

*☐ It need to be clearly stated that the years '2060' and '2063' are model years coming out of a not nudged simulation and are therefore relatively arbitrary. In some parts it sounds as if we get a really strong ozone loss in the future year 2063 and a really weak one in 2060. This needs to be clarified.*

*☐ The authors have included a lot of citations in their manuscript and compare their results with many of these studies. In some cases they need to be more specified. In my opinion some studies do not exactly show what is stated here.*

*☐ There are a lot of typos in the references. A cross check should be done before final publication.*

These points are addressed below.

*Specific comments:*
*☐ Page 4, line 27: How are orographic and non-orographic gravity waves parameterized? Please provide a reference.*

We have added this information to the manuscript.

*☐ Page 5, lines 15 - 17: I don't really understand why these six year bins are excluded. The supplement only shows which years are affected. Please provide some more information on this, here or in the supplement.*

For these periods there was an error when the model was run and we do not have sensible data.

*☐ Page 6, line 11: 'In all six reactions, a net loss of 2 odd oxygen molecules occurs per cycle.' → Do you mean 2 molecules ozone? But in cycle 3 there is only one. Please clarify this.*

We have clarified this in the manuscript.

Each cycle leads to net loss of 2 odd oxygen molecules ($O_x=O_3+O(^3P)+O(^1D)$). As discussed in Lee et al. (2002), the concentrations of ozone are significantly larger than of atomic oxygen, in particular in the polar lower stratosphere in winter/spring (which is the region studied here), therefore, we assume that $[O_x] \approx [O_3]$ and, thus, $d[O_x]/dt \approx d[O_3]/dt$.

☐ *Page 6, line 24: 'averaged from 65°N to the pole.' → Why do you use exactly 65°N 90°N? Have you also tested other latitudes, for example 60°N - 90°N and does this change the results?*

The 65-90°N average was chosen as a compromise between capturing a proportionally large fraction of the polar vortex while minimising the proportion of the extra vortex region, and including the edges of the vortex where in early winter halogen activation will take place preferentially (as there is little sunlight at higher latitudes).

☐ *Page 7, line 6: see comment above*

As above.

☐ *Page 10, lines 17 - 19: '...in agreement with Langematz et al. (2014).' → This statement should be specified. Do you compare with Figure 2a from Langematz et al. (2014)? From this figure I see a significant trend at 100 hPa, which is not the same as in your study. Moreover, you have to note that the time ranges are not identical.*

We have changed the text. ("In comparison, Langematz et al. (2014) found a statistically significant cooling trend in early winter over 1960-2100 throughout the polar stratosphere.")

☐ *Page 11, line 26: 'This is in broad agreement with the findings in Langematz et al. (2014).' → Where do you get this from? The focus in their study is on the vortex duration and not on the zonal wind trend. You need to be more specific with your comment.*

We have changed the text ("Langematz et al. (2014) analysed the timing of the formation of the NH polar vortex and found a statistically significant trend towards earlier vortex formation. It is possible that the strengthening of the stratospheric zonal wind in autumn/early winter in our ensembles could be related to a similar effect. ").

☐ *Page 12, line 3: Maybe you can call Section 3.3 'Case studies of exceptionally low and high ozone events' as you show results from both - low and high - and not only from low ozone events. This should be changed also in the Introduction (page 4, lines 5 - 7).*

We have changed the text to "Case study of exceptionally low and average ozone events".

☐ *Page 12, Section 3.3.: As you use free running, and no nudged model simulations, you won't expect that your 'model' years resemble 'real' years. Please make sure that the 'years' 2060 and 2063 are 'model' years. Do you really need this numbers? Maybe you can skip them and refer to low and high ozone events.*

We have added an explanatory sentence, and we also now refer to 'model years'.

☐ *Be very careful with the references on the figures:*

▪ *Page 12, lines 25 - 26: ...(see Fig. 7(b) and 8 (b)).*
▪ *Page 12, line 30: (Fig. 8(b)) and not 8(a)!!!*
▪ *Page 12, line 31: (Fig. 8(b)) and not 8(a)!!!*
▪ *Page 13, line 1: (Fig. 8(b)) and not 8(a)!!!*
▪ *Page 13, line 26: (Fig. 9(c)) and not 9(b)!!!*

Thank you for spotting this, we have corrected the text.

☐ *Page 15, line 31: '...account for ~20%...' → This is a very crude estimate. Please be more specific.*

We believe that this level of accuracy is adequate. Given the limitations of the diagnostics used, it is more appropriate to give an order of magnitude estimate than a precise (but not necessarily accurate) number.

*Technical corrections:*
☐ *Page 2, lines 9 - 10: reference for 'Montreal Protocol on Substances that Deplete the Ozone Layer'*

We have clarified that this is an international treaty.

☐ *Page 3, line 19: ... volume of PSCs (VPSC) → the abbreviation 'PSCs' has been introduced in line 9*

We have corrected the text.

☐ *Page 4, line 1: Stratosphere -troposphere Processes And their Role in Climate (SPARC)*

We have corrected the text.

☐ *Page 4, line 2: Please provide a reference for CCMI.*

We have added the requested reference.

☐ *Page 4, line 16: ... the recent SPARC Report on the Lifetimes of ... (SPARC, 2013;...) → Be careful that this is in line with the citation on page 21, line 14f.*

We have corrected the text.

☐ *Page 4, line 24: The dot at the end of the sentence is missing.*

We have corrected the text.

☐ *Page 5, line 13: You may introduce an abbreviation for 'sea-ice concentrations' here and use it on page 8, lines 18 and 27.*

We have added the abbreviation.

☐ *Page 5, line 17: ...long periods are excluded...*

It is the 'total' that 'is' excluded.

*Page 5, line 23: ... and a more minor ClO + O(3P) cycle... → You should include
(Cycle 3, reference) as before.*

We have added this.

 *Page 11, lines 3 - 4: The references should be sorted by year.*

We have corrected the text.

 *Page 11, lines 29 - 30: ...(see also Langematz et al., 2014).*

We have corrected the text.

 *Page 12, line 12: ... higher than in model year 2063.*

We have corrected the text.

 *Page 12, line 15: use the abbreviation 'BDC', as introduced before*

We have corrected the text.

 *Page 13, line 17: ... ClO concentrations in 2063 compared to 2060 (Fig. 9(c)). → The
figure shows a difference and not the concentrations in 2063.*

We have corrected the text. Also, we have added a figure showing the evolution of ClO at 21.5
km in the two case study years to the supplement.

 *Page 15, line 26: ...'exemplified by a case study in 2063.' → Either you include
'model year' here, or you skip the year. In the Conclusions I would prefer to skip the
years and use 'low and high ozone events instead.*

As suggested, we no longer use "2063" and "2060" in Sect. 4 (i.e. Conclusions).

 *Page 15, line 32: '...in year 2063 and a year from the same period ... ' → Better:
'...between this year and a year from the same period ...'*

Changed to "between this low ozone year and a year from the same period with near average
springtime ozone"'

 *Page 17, line 4: ... Steil, B.; and Tian, W....*

We have corrected the text.

*Page 17, line 5: The dot is missing at the end of the reference.*

We have corrected the text.

*Page 17, line 19: Drdla, K., and Müller, R.:...*

We have corrected the text.

*Page 19, line 10: ...Oberländer, S., ...*

We have corrected the text.

*Page 21, line 30: Tilmes, S., Müller, R., ...*

We have corrected the text.

*Page 22, line 5: ... and Müller, R.: ...*

We have corrected the text.

*Page 24, line 4: ... 11-year running average, respectively.*

We have corrected the text.

*Page 24, line 5:... 2060 and 2063, respectively, described in Sect. 3.3.*

We have corrected the text.

*Page 27, line 2: ... 11-year running average, respectively.*

We have corrected the text.

*Page 27, line 6: 'As in Figure 4, ...' → I would prefer an independent figure caption for Figure 6, as the only agreements with Figure 4 are the pressure levels and the meaning of the points and bars.*

We have added an independent caption for former Fig. 6 (now Fig. 7).

**REFERENCES**

Langematz, U., Meul, S., Grunow, K., Romanowsky, E., Oberländer, S., Abalichin, J., and Kubin, A.: Future Arctic temperature and ozone: The role of stratospheric composition changes, J. Geophys. Res.-Atmos., 119, 2092-2112, doi:10.1002/2013jd021100, 2014.

Lee, A. M., Jones, R. L., Kilbane-Dawe, I., and Pyle, J. A.: Diagnosing ozone loss in the extratropical lower stratosphere, J. Geophys. Res.-Atmos., 107, NO. D11, 4110, doi:10.1029/2001jd000538, 2002.

---

## Author Comment (AC2) · 29 Jun 2016

**AUTHORS' RESPONSE TO THE REFEREE 2 COMMENTS**

We thank Referee 2 for helpful comments regarding improving our manuscript. Below are point by point replies to the particular issues raised.

*General*
*The paper addresses an important scientific question, namely projections of future ozone levels in the Arctic. Overall the paper is well written and the discussions and arguments are clear (see below for some exceptions). The study uses a well established and well described model (the UM-UKCA model). The results on the timing of future Arctic ozone recovery are very relevant for the readership of ACP and also for future scientific ozone assessments. A particular strength of the paper is the use of ensembles and the analysis of relevant chemical and dynamical processes with a focus on the case of the simulation for winter 2063.*

*One major question to the models projecting the future is how well they simulate present day chemical Arctic ozone loss. Of course, this is a prerequisite for the assessment of future recovery. And it is not obvious that state-of-the-art models do a good job in all respects at present day chemical Arctic ozone loss. For example, Brakebusch et al. (2013) find a systematic high bias in ozone in the model of 18% in the lowermost stratosphere in March. They attribute most of this ozone bias to too little heterogeneous processing of halogens late in the winter and suggest that the model underpredicts ClONO2 early in the winter and has too little activated chlorine. How is the UM-UKCA model doing in this respect? How well does the model simulate denitrification (which is important for Arctic ozone loss)?*

The overall issue of model performance is addressed in the general authors' response. We have added further material to the manuscript that gives more detail on the overall performance of the model compared to observation/reanalysis data. We also note that all models exhibit biases. We also remind the reviewer that the objective here is to compare the UM-UKCA model for the present day with a period later this century, and the comparison is therefore internally consistent. In this case, model biases become somewhat less relevant.

*Further, as far as I understand the UM-UKCA model uses an equilibrium NAT scheme, where NAT is formed at the NAT equilibrium temperature, which likely overestimates the onset of heterogeneous reactivity in the model. Are there earlier studies, where these points have been addressed?*

Indeed, the UMUKCA simulates the NAT PSCs formation according to the equilibrium equation from Hanson and Mauersberger (1988; we now state it clearly in the manuscript). Yet, the use of equilibrium NAT scheme, although likely imperfect, is a common technique employed in CCMs (see e.g. Morgenstern et al., 2010).

*In the model heterogeneous reactivity is driven by NAT and ice particles, i.e., what regards the Arctic largely by NAT. This is likely not realistic, as there are extensive observations of liquid particles in the polar regions (e.g., Pitts et al., 2013). Nonetheless, Keeble et al. (2014) obtain a reasonable simulation of the Antarctic ozone hole assuming heterogeneous reactions on NAT and ice*

*using the model employed here. Similarly, Grooß et al. (2011) used a set-up with a NAT dominated heterogeneous chemistry (likely not realistic) but were able to reproduce the observed extremely low ozone values in the Antarctic. Possibly, it is not necessary to get every detail of PSC formation right to obtain a reasonable representation of chlorine activation and ozone loss in the model (see also Kirner et al., 2015; Solomon et al., 2015, and references therein). But I suggest that the issue of heterogeneous reactivity is discussed in more detail in the paper (see also detailed comments below).*

See below. We now acknowledge the relative simplicity of our scheme in Sect. 2.1, as well as rephrase the wording regarding the study of Keeble et al., 2014 (as suggested).

*Moreover, an important theme of the paper is halogen induced ozone loss due to heterogeneous reactions and chlorine activation (and the relative role of dynamics). As sufficiently cold conditions develop almost exclusively in the polar vortex, halogen induced ozone loss is only expected to occur in the vortex. However the analysis in the paper is mostly based on geographical latitude thereby neglecting the distinction between inside and outside of the vortex (in Fig. 7a however, a vortex average is presented, see also comments below). For example, how different would Figure 1 look, if equivalent latitude would be used rather than geographic latitude?*

Data at sufficiently high temporal resolution to convert diagnostics from geographical to equivalent latitudes are only available for one ensemble member. The use of only one ensemble member would limit our ability to explore the role of interannual variability, as is otherwise achieved by the use of ensemble simulations. Since the reviewer recognises the value of using a model ensemble for this study, we have retained the use of geographic latitudes. Note that we now give additional vortex averaged values for halogen induced ozone losses in the model case study years 2060/2063 in Sect 3.3.

*Further, in the discussions on the ozone anomaly simulated for the year 2063, the distinction between vortex processes and out of vortex processes is not always brought across clearly (see detailed comments below). Form my reading of the discussion in the paper (top of page 14), in the model, a significant fraction of the 2063 anomaly is driven by chemistry outside of the vortex – is this correct? I suggest improving the discussion and carefully quantify the contributions of chemistry and dynamics inside and outside of the polar vortex to the simulated ozone anomaly in 2063.*

These issues are addressed in general authors' response. As noted above, the vortex-average halogen induced ozone losses for the two case study years have been added to the manuscripts (Sect. 3.3). To avoid confusion, the 65-90°N passive ozone tracer diagnostic has been removed from the manuscript.

*In summary, I think with respect to several issues raised in this review, the paper needs to be revised and improved. Nonetheless, I believe that this is a potentially very good paper, which could make an important contribution to improved projections of future Arctic ozone levels and in particular regarding the various processes impacting polar ozone. The paper will also be very relevant*

*to the upcoming new WMO ozone assessment.*

*Detailed comments*
*• p 1, l 15: This statement is confusing: to me it implies that present day*
*spring Arctic ozone is 50-100 DU below the values expected after recovery*
*in 2060. Is this what you want to say here? Is this true for your model*
*simulations presented here?*

We have clarified the text. The sentence implies that the extreme low ozone episodes past 2060 have similar ozone levels as the average values, where no strong polar ozone depletion occurs, routinely simulated at the present day conditions (Fig. 1a in the manuscript).

*• p 1, l 20: why does an increase in downwelling lead to less consistency?*

We have changed the text to "…there is less confidence in the projected temperature trends in the lower stratosphere (100-50 hPa). This is partly due to an increase in downwelling …".

*• p 2, l 2: The use of CFCs did not lead to the 'suggestion…'*

We have corrected the text.

*• p. 2, l 4: One should distinguish the issue raised by Molina and Rowland*
*(1974) (upper stratospheric ozone, globally) from the ozone hole issue*
*pointed out by Farman et al. (1985).*

We do not think the text causes ambiguity.

*• p 2., l9: a citation from 1997 does not really allow to say 'soon' with*
*reference to Farman et al. (1985).*

We have changed it to 'later'.

*• p 3, l 1: the impact is also on ecosystems not only on human populations.*

We have added that to the text.

*• p 3, l 14: change 'sulphate' to 'cold sulphate'*

We have corrected the text.

*• p. 3, l 25: 'controlling' is perhaps to strong*

We have changed it to 'substantial'.

*• p 4, l 30: it might be worth pointing out that the mean age of the UMUKCA model is in*
*relatively good agreement with the observations in*
*the high latitudes of the Northern hemisphere (according to Chipperfield*
*et al., 2014), which is the most important region for this study.*

We have added the information about the UMUKCA age of air in the Northern Hemisphere high latitudes to the manuscript. We note it is more accurate to say here that the UMUKCA

mean age of air in the Arctic is at the lower end of the observationally derived values, as there is a spread of the mean age of air between different observational datasets (see SPARC, 2013).

*• I am assuming that not only this reaction is not taken into account on liquid aerosol but all five reactions listed in on page 13707 of Keeble et al. (2014). While this is likely not realistic (there are extensive observations of liquid particles in the polar regions, e.g., Pitts et al., 2013) is should not affect the quality of the ozone loss simulations too much. Assuming that the details of heterogeneous reactivity are not essential for a good representation of polar ozone loss; see also discussion below.*

As stated in the manuscript, the $ClONO_2$+HCl reaction is not included on liquid aerosols. The $HOCl$+HCl, $ClONO_2$+$H_2O$ and $N_2O5$+$H_2O$ reactions are included in the scheme. The only other reaction listed in Keeble et al. (2014) that is not included on liquid aerosols is the $N_2O_5$+HCl reaction, and we have now added this information to the manuscript.

*• p. 5, l 10-12: I do not think it is correct to say that Keeble et al. (2014) showed that ozone depletion can be attributed to heterogeneous reactions on NAT and ice. They obtain a reasonable simulation of the Antarctic ozone hole making this assumption. In the real world, for a long time, the heterogeneous reactivity will be dominated by ice particles. On the other hand, neglecting ice particles (and indeed NAT) does not result in a substantial change of the simulated ozone loss (Kirner et al., 2015; Solomon et al., 2015). So I think the wording should be more careful here.*

We agree with the reviewer and have clarified the text.

*• p 5, l 29: chemical formulas should not be in italics*

We have changed the formatting.

*• Section 2.3: I would suggest some more discussion of the relevance of the cycles discussed here. Could you roughly quantify what is meant with "lesser importance". It could be close to negligible for some of the cycles I think. On the other hand close to the tropopause in the non activated region natural (e.g. HOx driven cycles might be important for ozone loss.*

Some quantification of the contribution of individual cycles has been added to Sect.3.2.1. (see also our reply to the comment below).

*• Figure 1: how different would this figure look, if equivalent latitude would be used rather than geographic latitude. Would this not be the better choice? What is the reason for preferring geographic latitude vs. equivalent latitude?*

As explained above, the daily mean data needed to convert the diagnostics from geographical to equivalent latitudes are only available for one ensemble member. Using a single ensemble member would limit our ability to explore the role of interannual variability in our study.

*• p 7, l 15: change 'atmospheric' to 'stratospheric' – Cly is not defined in the troposphere*

We have corrected the text.

*• p 7, l. 20: I do not agree. The paper by Haigh and Pyle (1982) does not discuss the relevant ozone loss cycles in the polar regions, in particularly the ClO-dimer cycle, which does not slow down with decreasing temperature. I think you are discussing polar ozone loss in the lower stratosphere here.*

Whilst we agree that Haigh and Pyle (1982) do not explicitly discuss the NH polar regions, the increased ozone levels in the tropical mid/upper stratosphere driven by the GHG-induced cooling would then be transported by the BDC to the high latitudes, thereby contributing to increased ozone levels there.

Nonetheless, to avoid ambiguity, we have removed the sentence in question from the manuscript.

*• p 7, l 31: If I understand correctly, this value is computed by determining the minimum ozone value poleward of 65° N each day and then computing the mean value over a month. Have you ensured that all these values are within the polar vortex? Or could some of these values stem from (dynamically caused) so-called mini-holes?*

The minimum ozone values are determined from the monthly-mean data, and we believe the wording in the text is clear. As dynamically induced ozone mini-holes are likely to be short-lived, we believe it is unlikely that these would have a substantial impact on monthly mean values.

*• p 8, l 5: What is the implication of this statement? This sentence could be interpreted as stating that under present day conditions routinely strongly depleted ozone values are found. Please clarify.*

We have attempted to clarify the text by adding 'average' in front of 'values'. We believe it is clear that the word 'routinely' refers to average values, and does not refer to extremely low ozone events associated with the extremely cold Arctic winters.

*• P. 8, l 25-28: Difference to the results of Langematz et al. (2014); if the reason is 'differences in the representation', do you mean chemical or dynamical effects? If you agree with me that the difference is very likely not due to chemistry, you could state this point more clearly.*

We have changed 'due to differences in the representation' to 'due to differences, likely dynamical, in the representation ' in that sentence.

*• p 9, l 6: This is a bit misleading – are there more, even less important halogen cycles? I suggest stating which of the six cycles are dominant, which play a minor role and which are negligible.*

The wording has been corrected to 'six halogen cycles of most importance in the polar lower stratosphere'.

A general definition of the dominant and minor halogen cycles is in Sect. 2.3 where the halogen induced ozone loss diagnostic is defined. We have now added some extra information about the contribution of individual halogen cycles to the cumulative loss into Sect. 3.2.1.

*• p. 9, l 21: do you really mean 'halogen losses' here?*

Corrected to 'halogen induced ozone losses'. Also, we have added an explanatory sentence at the beginning of Sect. 3.2.1 stating that the terms 'halogen induced ozone loss' and 'halogen loss' are henceforth used interchangeably in reference to this diagnostic.

*• p 9, l 26: this formulation is a bit awkward; I think you never applied the 11-year running mean rather than removing it.*

We believe the formulation is correct. For clarity, we have changed '11-year running mean' to '11-year running ensemble mean' (both in the text and the caption to Fig. 3b).

*• p 10, l 3: not only the amount of PSCs also the length of the cold period. This is also important (Manney et al., 2011). Further below you also make this point.*

We have added this point the text.

*• p 10, l 18: change 'insignificant' to 'not significant'*

We have corrected the text.

*• p 10, l. 19: "Similar is true" – reformulate*

We have reformulated the sentence.

*• p 10, l 27-29: Actually, the ozone levels in Antarctica are also strongly influenced by the BD-circulation; it is just that the dynamical variability is lower – correct?*

We agree and have clarified the text.

*• p 11, line 1: provide a citation and/or explanation for w*

We have now added a citation and explanation to the text.

*• p 11, l 15: 'relatively' to what?*

Relative to the long-term trend, as well as to the range of variability found during the preceding and following periods.

*• p. 11, l 21-22: A central issue here is also the continued presence of PSCs and thus the continued activation.*

We agree and have added this point to the text.

*• p 11, l 32: will continue to occur in the future . . .*

We have corrected the text.

*• p 12, l 10: "long-term minimum of ensemble mean" – this is not quite clear? Which period is exactly considered? And what is meant is the lowest value for March mean ozone in the ensemble?*

We believe the sentence is clear. As stated in the text, the period considered is the late 1990s (We have now clarified this by replacing 'in the 1990s' with 'found in the 1990s'). We do not discuss 'the lowest value for March mean ozone in the ensemble' but the lowest value of the ensemble mean.

*• p 12, l 16: This effect could be reduced by considering equivalent latitude.*

We agree with the reviewer, but we have chosen to show vortex-averages to illustrate this instead. This approach also reduces the contribution of extra-vortex air, although we note the relative simplicity of the vortex definition used (see below).

*• p 12, l 17-19: Why did you choose 850 K to define the vortex? This is above the altitude where most halogen induced ozone loss occurs. Also how is the PV value defined (citation?)?*

Although the 850K level is above the main region of halogen induced ozone loss, it is a common measure for defining the polar vortex. We also note that our study examines both dynamical and chemical processes. Therefore, the choice of only a single level as representative of a polar vortex edge is, by definition, imperfect. The threshold is based on a rough estimate of where the maximum PV gradient occurs at this level. (See also figures R4-R5 in the general response that show stereographic maps of key quantities over the polar cap).

*• p 12, l 20: I suggest showing the vortex average data.*

As suggested by the reviewer, we have added the vortex-average total column ozone data for 2060 to Fig. 7a. (now Fig. 8a in revised manuscript)

*• p 13, l 9: this is an important point that should also be brought across clearly in the abstract.*

It was not clear to us which point on P13 L9 the reviewer was referring to. Since we do not give specific details of the model case study years in the abstract, we do not feel it would be fitting to add in a specific detail of this kind to the abstract. However, we emphasise that the increasing importance of dynamical processes for Arctic springtime ozone in the future is stated at the end of the abstract: "Whilst our results suggest that the relative role of dynamical processes for determining Arctic springtime ozone will increase in the future, halogen chemistry will remain a smaller but non-negligible contributor for many decades to come."

*• p 13, l 18: as stated before, heterogeneous reactivity in general should be more important than NAT formation in particular. Also, is there any formation of ice particles in the model for the year 2063?*

As discussed in Sect. 2.1 of the updated manuscript, the heterogeneous reactivity in the NH polar regions in these experiments is likely to be dominated by NAT PSCs.

The November-March mean volume of ice (1-25 km) in the model year 2063 is ~27×10$^6$ km$^3$.

*• p 14, l 2-4: it is interesting to note that only part of the effect of the anomaly has its origin in processes in the polar vortex. Doesn't this mean that in the model only part of the chemistry driven effect is caused by halogen chemistry? Again I suggest to bring this message more clearly across the the abstract.*

These issues are addressed in general authors' response (see also below).

We note that the chemical ozone loss diagnostic derived from the passive ozone tracer results from a complex balance between chemical loss and production cycles as well as their interaction with transport throughout the winter.

To avoid confusion, we now state it clearly in the text; we have also removed the 65-90°N passive ozone tracer diagnostic from the text (and replaced with vortex-average quantity in the former Fig. 7a (now 8a)), as well as reformulated parts of the last two paragraphs of Sect. 3.3. Lastly, the vortex-average halogen induced ozone losses for the two case study years have been added to the manuscript (Sect. 3.3).

*• p 14, l 9: here you state that the halogen effect is 40 DU but above you state that the polar vortex effect is only 25 DU. Does this mean that in the model, a significant fraction of the 2063 anomaly is driven by halogen chemistry outside of the vortex? Is there chlorine activation outside of the vortex in the model? This discussion at this stage and the attribution of ozone loss to processes needs to be improved.*

As above, these issues are addressed in the general authors' response.

In the original version of the manuscript we state that, indeed, halogen induced ozone loss in the 1-25 km layer is ~40 DU. The chemical loss, as derived from the passive ozone tracer, due to all cycles integrated from the surface to the top of the atmosphere, vortex-averaged, is ~25 DU. Note that the two diagnostics are not equivalent. The chemical ozone loss diagnostic derived from the passive ozone tracer results from a complex balance between chemical loss and production cycles as well as their interaction with transport. To avoid confusion, we now state it clearly in the text, as well as modify the manuscript as described above and in the general authors' response.

Note also that the vortex-average halogen induced ozone losses for the two case study years (now added to the manuscript, but see also the general authors' response), show mostly similar or somewhat higher values that the Arctic mean values.

*• p 14, l 21: citation for 'other studies'*

We have added the requested citation.

*• p 15, l 31: it is not clear to me where the number of 20% is coming from. on p. 14, you report that the halogen induced loss in 2063 is twice that of 2060, which might be the first order information of interest here. However, to me the question is still open of how much the difference between 2060 and 2063 is a polar vortex effect and in how far is is influenced substantially by out of vortex processes.*

The difference in the 65-90°N halogen induced ozone loss between the two model years is ~20 DU (we have added this to the manuscript). The difference in the total ozone column by the end of March between the two years is ~100 DU. Hence, the difference in the estimated halogen loss in the polar lower stratosphere equates with ~20% of the full ozone difference. We have now changed 'can therefore account for' to 'is equivalent to'.

Vortex-average halogen induced ozone losses for the two case study years have been added to the updated manuscript.

See also the general authors' response.

**REFERENCES**

Haigh, J. D., and Pyle, J. A.: Ozone perturbation experiments in a two-dimensional circulation model, Q. J. Roy. Meteor. Soc., 108, 551-574, 10.1002/qj.49710845705, 1982.

Keeble, J., Braesicke, P., Abraham, N. L., Roscoe, H. K., and Pyle, J. A.: The impact of polar stratospheric ozone loss on Southern Hemisphere stratospheric circulation and climate, Atmos. Chem. Phys., 14, 13705-13717, doi:10.5194/acp-14-13705-2014, 2014.

SPARC: SPARC Report on the Lifetimes of Stratospheric Ozone-Depleting Substances, Their Replacements, and Related Species, SPARC Report No. 6, 2013.

Morgenstern, O., et al., Review of the formulation of present-generation stratospheric chemistry-climate models and associated external forcings, J. Geophys. Res., 115, D00M02, doi:10.1029/2009JD013728, 2010.

Hanson, D. and Mauersberger, K.: Laboratory studies of the nitric acid trihydrate - Implications for the south polar stratosphere, Geophys. Res. Lett., 15, 855-858, 1988.

---

## Author Comment (AC3) · 29 Jun 2016

**AUTHORS' RESPONSE TO THE REFEREE 3 COMMENTS**

We thank Referee 3 for helpful comments regarding improving our manuscript. Below are point by point replies to the particular issues raised.

*This article is based on an interesting study of the projection of future Arctic ozone using ensemble simulation from the UM-UKCA chemistry climate model. While other studies have been performed on this subject (e.g. WMO, 2011; Langematz et al., 2014), the originality of the study lies in the use of ensemble simulation, which allows the authors to estimate the intrinsic variability of the stratosphere, together with the impact of ozone depleting substances decrease and climate change on Arctic ozone. The paper is well written and informative for the projection of future Arctic ozone and I recommend publication in ACP, provided that important comments for improvement are taken into account.*

*Main comments*
*The main focus of the study is on the respective contribution of chemistry and dynamics on future Arctic ozone. In that respect diagnostics have been set up in order to evaluate the importance of chlorine chemistry in future ozone loss and the authors argue that halogen chemistry can still play a substantial role after mid-century. Since this result is rather intriguing, it deserves more attention in the article. A whole section is dedicated to the case study of winter 2063 but it is somewhat descriptive and does not demonstrate fully that the chemical loss is linked to halogen chemistry. For example, is the observed loss coherent with the known relationship between Cly levels, chlorine activation and PSC volume (e.g. Rex et al., 2004)?*

An analysis of potential Vpsc vs. halogen induced ozone loss has been added to the updated manuscript (Sect. 3.2.3), where we also compare the model results to the study by Rex et al. (2004, 2006). As illustrated by the red star in Fig. 5 in the manuscript, the relationship between potential Vpsc and halogen loss simulated in the case study model year 2063 compares well with the fit to the ensemble data for that period. Also, we have added Fig. S3 shown below to the supplementary material to illustrate the evolution of Arctic mean ClO, $Cl_2O_2$, HCl and $ClONO_2$ for the two model case study years.

[Figure]

Figure S3. Timeseries of 65-90°N daily mean ClO and $Cl_2O_2$ [ppt] (left) and HCl and $ClONO_2$ [ppb] (right) at 21.5 km for the case study model years 2063 (solid lines) and 2060 (dashed lines).

*What is the role of nitrogen chemistry that can sometimes be important in the Arctic mid-stratosphere as shown in Kuttipurath et al., 2010?*

We have estimated the 65-90°N cumulative (1.Nov-30.Mar) ozone loss in the lower atmosphere (1-25 km) due to the $NO_2 + O(^3P)$ reaction to be ~2DU/4DU in the model years 2063/2060, respectively.

While we agree that all ozone loss cycles ($HO_x$, $NO_x$… etc) are important for the evolution of ozone (we now emphasize this in later part of Sect. 3.3), especially outside of the polar vortex and/or in the mid-/upper stratosphere, the focus of our study is on ozone losses due to halogen chemistry in the lower stratosphere. To avoid confusion we have removed the 65-90°N average chemical ozone loss diagnostic from the manuscript, leaving only the vortex-average quantity.

*A quantification of PSC volume and a figure similar to Figure 2 but showing observations in order to demonstrate the skills of the model to simulate halogen chemistry would be useful.*

An analysis of potential Vpsc vs. halogen induced ozone loss in the model has been added to the updated manuscript (Sect. 3.2.3), where we have made a comparison to the studies of Rex et al. (2004; 2006). While a comparison of the modelled total ozone column with observations was presented in Fig. 1(a), we have now added additional material (Sect. 2.1) that discusses present day ozone/ClO from a similar version of the model with 'nudged' meteorology and from observations (see the general authors' response). For the future period, it is of course impossible to compare our model results to observations.

*Since the Arctic ozone loss is computed over the 65-90° N latitude range, it encompasses some loss from non-vortex air. This issue is acknowledged by the authors but would need some quantification.*

The estimated vortex average halogen induced ozone losses for the two case study years 2063/2060 have been added to the updated manuscript (last paragraph in Sect. 3.3). See also the general authors' response.

*From Figure 1, it seems that the interannual variability of Arctic ozone from the ensemble simulation is larger than the natural variability as seen from the observations. Can the authors comment on that and provide some statistics on this issue?*

We do not agree. The variability in the observed ozone column appears comparable to or even somewhat larger than in our model.

*In addition a more substantial description in section 2 of the skills of the UM-UKCA model in terms of polar ozone simulation is needed: e.g is there a cold bias of the polar stratosphere? How the strength and duration of the Northern vortex compare with observations, . . .?*

A comparison of Northern hemisphere climatological stratospheric zonal wind and temperatures with reanalysis data has been added to the revised manuscript (Sect. 2.2). We have also added a comparison of present day polar ozone/ClO for a "nudged" meteorology version of UMUKCA with satellite observations.

See also the general authors' response.

*Temperature trends: No mention is made of the evolution of the occurrence of sudden warmings in the ensemble simulation. It is thus difficult to distinguish radiatively induced with dynamically induced temperature trends. This issue should be addressed.*

While we acknowledge that changes in the frequency of sudden stratospheric warmings (SSWs) may be important for determining polar temperature trends, a quantitative distinction between the dynamically and radiatively-induced temperature trends is beyond the scope of the study. In addition, data at sufficiently high temporal resolution (i.e. daily) required to calculate SSW occurrences are only available for one ensemble member, and this is unlikely to be adequate for diagnosing statistically robust changes in SSW frequency.
We have now changed the sentence 'The ensemble shows a radiatively-driven cooling trend…' in the abstract to 'The ensemble shows a significant cooling trend…'.

*Minor comments*
*P5 l24: it is not clear how the 11-year solar cycle is simulated over the 21st century.*

Solar cycle variability for the future period (after 2009) is included as in earlier periods but with a repeating sinusoidal 11-year cycle with an amplitude derived from observed cycle 23 (see Jones et al., 2011; Gray et al., 2013). We have added this information to the revised manuscript.

*P7 l21: the products of the reaction are wrong: it should be Cl + O2.*

Thank you for spotting this, but note that this has already been corrected after the initial quick referees' reviews and prior to the publication in ACPD.

*P6 l4: This sentence is not so clear. The computational efficiency relates to the use of the diagnostics for evaluating halogen induced ozone loss.*

We have improved the language by replacing 'computational efficiency' with 'computational ease'. Also, since we now include additional vortex-average halogen induced ozone loss diagnostics (see above and the general authors' response), we have added "…(except for the vortex-averaged quantities reported in Sect. 3.3, where daily means are used)…" to the sentence.

*P9 l21: what is the contribution of slowing of gaz phase ozone loss cycles compared to changes in stratospheric transport in the earlier recovery of Arctic ozone?*

Referee 2 also questions this point. We have removed this sentence as it is clearly confuses the reader.

*P11 l4-13: In line with my major comments, the causes of the drops of Arctic ozone in late century, and the comparison with Langematz et al. (2014) study should be better substantiated.*

We have now added more quantitative material, relating, e.g., potential Vpsc to halogen induced ozone loss; plus evolution of ClO$_x$ and chlorine reservoirs for the model years 2063/2060 (see Figure S3).

*P12 l10-18: a chemical loss of 40 DU is similar to the current Arctic ozone losses, while chlorine levels in 2061-2080 will be lower by more than a factor of 2. What PSC volume is necessary for such extreme loss?*

The potential Vpsc calculated for the model case study year 2063 is now shown as the red star in Fig. 5 of the revised manuscript, and some text referencing to this is in Sect. 3.3.

*P16 l13: what is the justification for the PV value to define the vortex?*

This is a simple and fairy arbitrary choice based on a rough estimate of the maximum PV gradient. We are simply trying to be illustrative here. A similar approach has been used in other studies e.g. Müller et al (2005).

*Figure 5: Case study years (2060 and 2063) should be highlighted in the figure.*

We have added these as points to the figure (now Fig. 6).

**REFERENCES**:

Rex, M., Salawitch, R. J., von der Gathen, P., Harris, N. R. P., Chipperfield, M. P., and Naujokat, B.: Arctic ozone loss and climate change, Geophys. Res. Lett., 31, L04116, doi:10.1029/2003gl018844, 2004.

Rex, M., Salawitch, R. J., Deckelmann, H., von der Gathen, P., Harris, N. R. P., Chipperfield, M. P., Naujokat, B., Reimer, E., Allaart, M., Andersen, S. B., Bevilacqua, R., Braathen, G. O., Claude, H., Davies, J., De Backer, H., Dier, H., Dorokhov, V., Fast, H., Gerding, M., Godin-Beekmann, S., Hoppel, K., Johnson, B., Kyro, E., Litynska, Z., Moore, D., Nakane, H., Parrondo, M. C., Risley, A. D., Skrivankova, P., Stubi, R., Viatte, P., Yushkov, V., and Zerefos, C.: Arctic winter 2005: Implications for stratospheric ozone loss and climate change, Geophys. Res. Lett., 33, L23808, doi:10.1029/2006gl026731, 2006.

Gray, L. J., Scaife, A. A., Mitchell, D. M., Osprey, S., Ineson, S., Hardiman, S., Butchart, N., Knight, J., Sutton, R., and Kodera, K.:, A lagged response to the 11 year solar cycle in observed winter Atlantic/European weather patterns, J. Geophys. Res. Atmos., 118, 13,405–13,420, doi:10.1002/2013JD020062, 2013.

Jones, C. D., Hughes, J. K., Bellouin, N., Hardiman, S. C., Jones, G. S., Knight, J., Liddicoat, S., O'Connor, F. M., Andres, R. J., Bell, C., Boo, K. O., Bozzo, A., Butchart, N., Cadule, P., Corbin, K.

D., Doutriaux-Boucher, M., Friedlingstein, P., Gornall, J., Gray, L., Halloran, P. R., Hurtt, G., Ingram, W. J., Lamarque, J. F., Law, R. M., Meinshausen, M., Osprey, S., Palin, E. J., Chini, L. P., Raddatz, T., Sanderson, M. G., Sellar, A. A., Schurer, A., Valdes, P., Wood, N., Woodward, S., Yoshioka, M., and Zerroukat, M.: The HadGEM2-ES implementation of CMIP5 centennial simulations, Geosci. Model Dev., 4, 543-570, doi:10.5194/gmd-4-543-2011, 2011

R. Müller, S. Tilmes, P. Konopka, J.-U. Grooß, H.-J. Jost: Impact of mixing and chemical change on ozone-tracer relations in the polar vortex, Atmospheric Chemistry and Physics, European Geosciences Union, 2005, 5 (11), pp.3139-3151

---

## Author Comment (AC4) · 29 Jun 2016

**Authors' general response**

We thank the three anonymous Reviewers for their helpful comments regarding improving our manuscript. We reply to the individual comments in the separate responses. In addition, Reviewers 2 and 3 raised a couple of common questions. These are in turn addressed below.

**1. How well does the UMUKCA model simulate the present day Arctic ozone loss?**

This will depend both on how well the model's chemistry scheme performs and how well the model reproduces the Arctic meteorology. These two factors are addressed below.

The performance of the UMUKCA CheS+ chemistry scheme can be evaluated using a similar model version but with a so-called specified dynamics set-up, i.e. in which model's temperature and winds are nudged towards meteorological reanalysis data. A comparison between the Arctic/Antarctic mean (65-90°N/S) total ozone columns in this nudged CCMI REFC1 CheS+(SD) integration[1] shows a good correlation between the simulated and observed ozone (Bodeker et al. 2005; Müller et al., 2008) (see Fig. S1). In the Antarctic, there is some overestimation of the ozone column in the model by up to ~15-30 DU (see Fig. S1).

For the cold March 2011, the model shows a positive bias in the zonal mean monthly mean ozone levels in the Arctic lower stratosphere (30-100 hPa) of up to ~0.9 ppb (~41%; for the 70-90°N mean, 50 hPa) compared with MIPAS satellite data (Fisher et al., 2008, Fig. R1). The positive ozone bias is commensurate with a negative bias in the zonal mean monthly mean ClO levels at this altitude (not shown). Note that there is some uncertainty in the observed zonal mean monthly mean ClO, in part due to sparse temporal sampling (twice a day) of the MIPAS instrument.

In general, all models will show some biases with respect to observations. Importantly, as in many studies, we compare the model for the present day and future periods in an internally consistent way; therefore, any biases will become less relevant for our study.

Regarding the NAT PSC formation and removal, as described in Chipperfield et al. (1999), the NAT PSC formation follows the equilibrium expression from Hanson and Mauersberger (1988). Although potentially imperfect, the use of equilibrium NAT scheme is a common technique employed in CCMs (see e.g. Morgenstern et al., 2010). The denitrification scheme assumes a relatively slow NAT sedimentation velocity of ~40 m/day for pure NAT PSCs, and a much faster NAT sedimentation velocity of ~1540 m/day in the presence of ice; the latter assumes coating of NAT PSCs onto ice particles.

Parts of the above information have been added to Sect. 2.1 of the revised manuscript. Figure S1 has also been added to Supplementary Information.
* * *
[1]We thank Paul Telford for preparing and running the nudged UMUKCA CCMI REFC1 CheS+(SD) integration discussed above (see also the updated Manuscript for full list of acknowledgements).

[Figure]

[Figure]

*Figure S1. The evolution of 1979-2012 65-90°N March (top) and 1979-2011 65-90°S October (bottom) total ozone column [DU] in the nudged UMUKCA CCMI REFC1 CheS+(SD) integration (red) and observations (black, Bodeker total ozone column dataset: Bodeker et al., 2005; Müller et al., 2008).*

[Figure]

*Figure R1. Monthly mean 70-90°N March ozone [ppm] mixing ratios in individual years from 2006 to 2011. Red lines are for the nudged UMUKCA CCMI REFC1 CheS+(SD) simulation and black lines are for MIPAS data (Fisher et al., 2008). Thick lines highlight the year 2011.*

Beside chemistry, winter/springtime Arctic ozone levels are also strongly controlled by meteorology; any biases in temperature and winds will therefore impact on the simulated ozone. The UMUKCA REFC2 ensemble integrations described in the manuscript show a somewhat weaker and warmer present day Arctic stratospheric vortex in early/mid-winter, with a slight zonal wind bias of up to ~6 ms$^{-1}$ in the mid-latitude lower/mid- stratosphere in March (see Fig. R2 and R3 of this response).

This information has been included in Sect. 2.2 of the updated manuscript.

As discussed in Sect. 3.2.3 of the updated manuscript, the model captures qualitatively the diagnosed relationship that higher Vpsc is associated with higher halogen induced ozone losses (Rex et al., 2004, 2006). The modelled correlation is R≈0.8 for the individual 20 year periods in the 21st century, demonstrating the coupling between chemistry and meteorology.

[Figure]

*Figure R2. Contours: November to April monthly mean zonal mean zonal wind [m/s] climatology over 1979-2010 in the mean of ensemble members 1 and 2. Shading: the difference [m/s] between the model and the ERAI reanalysis (Dee et al., 2011).*

[Figure]

*Figure R3. As in Fig. R2 but for temperature [K].*

2. Chemical losses inside and outside polar vortex.

As stated in the manuscript, the model's passive ozone tracer does not participate in any chemical processing, and not just that due to halogen chemistry. This includes gas phase chemistry that is particularly important outside the polar vortex, as well as heterogeneous and gas phase chemistry inside and at the edge of the polar vortex. To avoid confusion with the interpretation of the passive ozone tracer diagnostic, we have replaced the Arctic mean passive ozone tracer with the vortex-averaged tracer in Fig. 8(a) of the revised manuscript. In addition, we have removed the last two sentences of Sect. 2.3 and the polar cap quantity from the last but one paragraph of Sect. 3.3. Lastly, we have reformulated the passive ozone paragraph in Sect. 3.3., including a cautionary sentence regarding the interpretation of the difference between the passive and full ozone column as resulting of a complex balance between chemical loss and production cycles as well as their interaction with transport throughout the winter.

In comparison, the halogen induced ozone loss diagnostic (Sect. 2.3) inside the polar vortex (calculated using various definitions of the polar vortex edge) lower atmosphere shows that the losses for model years 2063/2060 are mostly similar or even somewhat higher than those calculated for the Arctic mean (See Table 1 as well as Fig. R4-R5 below). This supports the use and interpretation of the Arctic mean diagnostic for analysing the long-term trends and variability in the halogen induced ozone loss for the full ensemble (Sect. 3.2.1).

The vortex-averaged halogen induced ozone losses for the two case study years have now been added to the manuscript (Sect. 3.3). We stress that we use a simple definition of the polar vortex edge, based on a rough estimate of a characteristic PV value for the maximum PV gradient.

CUMULATIVE HALOGEN-INDUCED OZONE LOSS [DU]

|  | 65-90°N | $PV_{850K} \geq 600$ PVU | $PV_{450K} \geq 30$ PVU | $PV_{450K} \geq 35$ PVU |
|---|---|---|---|---|
| 2063 | 39 | 44 | 46 | 43 |
| 2060 | 18 | 23 | 23 | 6 |

*Table 1. Cumulative halogen induced ozone loss (1 Nov-30 Mar, 1-25 km) for the two case study years 2063 and 2060, calculated for the 65-90°N mean and a number of polar vortex edge definitions.*

[Figure]

*Figure R4. Daily mean total ozone column [DU] (top left), temperature at 21.5 km [K] (top right), halogen induced ozone loss in the 1-25 km layer [DU/day] (bottom left) and ClO [ppt] (bottom right) at 21.5 km simulated on 1 March in the case study year 2060.*

[Figure]

*Figure R5. As in Fig. R4 but for the case study model year 2063.*

**References:**

Chipperfield, M. P.: Multiannual simulations with a three-dimensional chemical transport model, J. Geophys. Res.-Atmos., 104, 1781-1805, doi:10.1029/98jd02597, 1999.

Dee, D. P., Uppala, S. M., Simmons, A. J., Berrisford, P., Poli, P., Kobayashi, S., Andrae, U., Balmaseda, M. A., Balsamo, G., Bauer, P., Bechtold, P., Beljaars, A. C. M., van de Berg, L., Bidlot, J., Bormann, N., Delsol, C., Dragani, R., Fuentes, M., Geer, A. J., Haimberger, L., Healy, S. B., Hersbach, H., Hólm, E. V., Isaksen, L., Kållberg, P., Köhler, M., Matricardi, M., McNally, A. P., Monge-Sanz, B. M., Morcrette, J.-J., Park, B.-K., Peubey, C., de Rosnay, P., Tavolato, C., Thépaut, J.-N. and Vitart, F., The ERA-Interim reanalysis: configuration and performance of the data assimilation system. Q.J.R. Meteorol. Soc., 137: 553–597. doi: 10.1002/qj.828, 2011.

Fischer, H., Birk, M., Blom, C., Carli, B., Carlotti, M., von Clarmann, T., Delbouille, L., Dudhia, A., Ehhalt, D., Endemann, M., Flaud, J. M., Gessner, R., Kleinert, A., Koopman, R., Langen, J., López-Puertas, M., Mosner, P., Nett, H., Oelhaf, H., Perron, G., Remedios, J., Ridolfi, M., Stiller, G., and Zander, R.: MIPAS: an instrument for atmospheric and climate research, Atmos. Chem. Phys., 8, 2151-2188, doi:10.5194/acp-8-2151-2008, 2008.

Rex, M., Salawitch, R. J., von der Gathen, P., Harris, N. R. P., Chipperfield, M. P., and Naujokat, B.: Arctic ozone loss and climate change, Geophys. Res. Lett., 31, L04116, doi:10.1029/2003gl018844, 2004.

Rex, M., Salawitch, R. J., Deckelmann, H., von der Gathen, P., Harris, N. R. P., Chipperfield, M. P., Naujokat, B., Reimer, E., Allaart, M., Andersen, S. B., Bevilacqua, R., Braathen, G. O., Claude, H., Davies, J., De Backer, H., Dier, H., Dorokhov, V., Fast, H., Gerding, M., Godin-Beekmann, S., Hoppel, K., Johnson, B., Kyro, E., Litynska, Z., Moore, D., Nakane, H., Parrondo, M. C., Risley, A. D., Skrivankova, P., Stubi, R., Viatte, P., Yushkov, V., and Zerefos, C.: Arctic winter 2005: Implications for stratospheric ozone loss and climate change, Geophys. Res. Lett., 33, L23808, doi:10.1029/2006gl026731, 2006.

Bodeker, G. E., Shiona, H., and Eskes, H.: Indicators of Antarctic ozone depletion, Atmos. Chem. Phys., 5, 2603-2615, 2005.

Müller, R., Grooss, J. U., Lemmen, C., Heinze, D., Dameris, M., and Bodeker, G.: Simple measures of ozone depletion in the polar stratosphere, Atmos. Chem. Phys., 8, 251-264, 2008.

Hanson, D. and Mauersberger, K.: Laboratory studies of the nitric acid trihydrate - Implications for the south polar stratosphere, Geophys. Res. Lett., 15, 855-858, 1988.

Morgenstern, O., et al., Review of the formulation of present-generation stratospheric chemistry-climate models and associated external forcings, J. Geophys. Res., 115, D00M02, doi:10.1029/2009JD013728, 2010.

---

## Author Comment (AC5) · 29 Jun 2016

[revised manuscript text omitted]

**(b) temperature at ~22.7 km**

[Figure]

**(a) total ozone column**

[Figure]

**(b) temperature at ~22.7 km**

[Figure]

Figure 78. Timeseries of Northern Hemisphere (a) daily total column ozone [DU] during the winters 2060 (black) and 2063 (red), and (b) temperature [K] at 22.7 km. Solid lines show the Arctic mean (65-90°N), dashed red lines in (a) shows polar vortex averages for 2063 (see text for details) and dotted lines in (b) show minimum daily mean temperatures found anywhere poleward of 65°N. The blue dashedsolid line in (a) shows the evolution of the vortex-averagedArctic mean passive ozone tracer in 2063.

[Figure]

Figure 89. Time-altitude cross-sections of the daily Arctic (60°N) zonal mean zonal wind [ms$^{-1}$] in (a) 2060 and (b) 2063.

[Figure]

Figure 910. Colours: Time-altitude cross-sections of the Arctic mean differences between the years 2063 and 2060 in (a) ozone mixing ratios [ppm], (b) ozone number density [$10^{12}$ molecules cm$^{-3}$] and (c) ClO mixing ratio [ppt]. The solid contours show the year 2060 for reference.

---

## Author Comment (AC6) · 29 Jun 2016

As noted in Sect. 2.2, of the manuscript, our UMUKCA REFC2 ensemble of integrations consists of 2 full 1960-2099 integrations (ENS1-2) and 5 shorter runs from covering November 1980 to December 2080 (ENS3-7). For technical reasons, data from 5 six-year-long intervals in total were excluded from the analysis, in particular:

- July 2025 – June 2031 in member number 5 (ENS5)
- April 2074 – March 2080 in member number 5 (ENS5)
- April 1996 – March 2002 in member number 6 (ENS6)
- April 2043 – March 2049 in member number 6 (ENS6)
- August 1982 – July 1988 in member number 7 (ENS7)

An example of the resulting timeseries is shown in Fig. S2₁ for 65-90ºN March total ozone column.

total ozone column, 65-90°N, March

[Figure]

total ozone column, 65-90°S, October

[Figure]

Figure S1. The evolution of total ozone column [DU] over 1979-2012 for 65-90°N March (top) and over 1979-2011 for 65-90°S October (bottom) in the nudged UMUKCA CCMI REFC1 CheS+(SD) integration (red) and observations (black, Bodeker total ozone column dataset: Bodeker et al., 2005; Müller et al., 2008). See Sect. 2.1 for details.

5

[Figure]

Figure S2̶1̶.: (a-g) Timeseries of 65-90°N March total column ozone ̶c̶o̶l̶u̶m̶n̶s̶ [DU] for individual ensemble members as labelled (black). Red lines show the corresponding ̶l̶o̶n̶g̶ ̶t̶e̶r̶m̶ linear trends over the 2000-2080 period.

[Figure]

Figure S3. Timeseries of 65-90°N daily mean ClO and $Cl_2O_2$ [ppt] (left) and HCl and $ClONO_2$ [ppb] (right) at 21.5 km for the model case study years 2063 (solid lines) and 2060 (dashed lines).

---

## Referee Report (RR1)

Review of  revised version of

**"Future Arctic ozone recovery: the importance of chemistry and dynamics"**

by E. M. Bednarz

**General Comments**

The authors have done a very through job revising their paper. They have taken great care of taking all the comments by all the reviewers into account. I have a few remaining comments, partly also on newly added material that I would suggest the authors take into account when revising their paper for publication in ACP.

The authors have added an analysis of the halogen induced ozone loss in the model vs VPSC (new Fig. 5). I think this is a valuable addition to the paper. The authors obtain different slopes of VPSC against chemical ozone loss for different time periods due to changes in the stratospheric halogen loading (as expected). There have been attempts to construct a variant of VPSC (PACL, Tilmes et al., 2007) than includes the changing halogen loading of the stratosphere in the definition. This quantity has been applied to an evaluation of heterogeneous processes in the polar lower stratosphere in the Whole Atmosphere Community Climate Model (WACCM3) for the period 1960—2003. It would be interesting to see the behaviour of the simulated ozone loss against PACL and see if the different slopes would collapse onto one curve. It would be good to add this aspect to the paper.

There was also a lot of discussion in the reviews about the representation of polar chemistry in general and heterogeneous processes in particular in the UM-UKCA model. The authors have explained now their equilibrium NAT scheme, where NAT is formed at the NAT equilibrium temperature, which I think is acceptable for this paper. However, it is known since a long time that NAT does *not form* at equilibrium so I would argue it should no longer be a "common technique employed in CCMs". An aspect that I did not mention in the first review is the parametrization for reactivity on NAT that is used in the model. There are two competing and rather different formulations for NAT reactivity (Carslaw et al., 1997a,b); as NAT is important in the heterogeneous chemistry used here it might be worth clarifying this point in the paper.

Further, the heterogeneous reaction $HCl + ClONO_2$ is not included on liquid surfaces. But it is well known to occur  on liquid surfaces (e.g., Hanson et al., 1994), so what is the reason for not including it? Possibly, the effects of a too early onset of NAT chemistry and the neglect of $HCl + ClONO_2$ on liquid surfaces cancel out to some extent. There is no point in changing anything here with regard to the manuscript in question. However, I suggest correcting these two issues in future model versions, i.e. include $HCl + ClONO_2$ on liquid surfaces and introduce a supersaturation requirement for NAT formation.

The details of the heterogeneous chemistry could however matter for predictions of the future; for example it is stated in the paper now that :"Stratospheric $H_2O$ and $HNO_3$ levels are projected to increase in the future, which is likely to enhance levels of PSCs" However it depends on the type of PSC in question in how far an increase of $H_2O$ or $HNO_3$ will affect chlorine activation. Nonetheless, the halogen induced polar ozone loss in the model will be in many respects robust against assumptions on heterogeneous chemistry (e.g. NAT vs. liquid; Solomon et al., 2015, see also discussion and references in my previous review). As this aspect is relevant for the model results presented here, I suggest discussing the issue briefly in the paper. For example on p 5, l. 5 you could briefly state not only that Keeble et al. (2014) obtained a reasonable representation of springtime Antarctic ozone, but also why. Perhaps something along these lines ~ "Previous studies (refs) obtain a consistent and relatively realistic representation of springtime Antarctic ozone for rather different assumptions on PSC formation so that the details of the heterogeneous chemistry in the model runs presented here should have no strong impact on the simulated halogen induced ozone loss. Moreover, we compare here the time evolution of polar ozone in a consistent framework, so that possible model biases should be less important". Just a suggestion.

Finally, an important point in the reviews was also the discussion of chemical losses inside and outside the polar vortex. And a comparison between the 2060 and the 2063 model case. I see that the authors have responded in detail to this point and have modified and improved the manuscript accordingly. I think the analysis conducted for this discussion would also be very helpful to the reader (and not only to the reviewer/editor) and should be added to the manuscript. Perhaps in the supplement, but I suggest adding table 1 and Figs. R4 and R5 (or at least the left-hand columns of R4 and R5) to the actual paper together with the accompanying discussion from the reply (which then needs perhaps be a bit

extended/reformulated). But the material is there and I would recommend showing it.

In summary, I think the authors have done a very careful job in replying to the reviews and in revising the paper. I believe some further improvements are possible as outlined above. With these modifications, which should not be too difficult to do, I recommend the paper for publication in ACP.

**References**

Carslaw, K. S. and Peter, T.: Uncertainties in reactive uptake coefficients for solid stratospheric particles – 1. Surface chemistry, Geophys. Res. Lett., 24, 1743–1746, 1997a.

Carslaw, K. S., Peter, T., and Müller, R.: Uncertainties in reactive uptake coefficients for solid stratospheric particles – 2. effect on ozone depletion, Geophys. Res. Lett., 24, 1747–1750, 1997b.

Hanson, D. R., Ravishankara, A. R., and Solomon, S.: Heterogeneous reactions in sulfuric acid aerosols: A framework for model calculations, J. Geophys. Res., 99, 3615–3629, 1994.

Solomon, S., Kinnison, D., Bandoro, J., and Garcia, R.: Simulation of polar ozone depletion: An update, J. Geophys. Res.-Atmos., 120, 7958-7974, 10.1002/2015jd023365, 2015.

Tilmes, S., Kinnison, D., Müller, R., Sassi, F., Marsh, D., Boville, B., and Garcia, R.: Evaluation of heterogeneous processes in the polar lower stratosphere in the Whole Atmosphere Community Climate Model, J. Geophys. Res., 112, D24301, doi:10.1029/2006JD008334, 2007.

**Technical issues**

Throughout the paper: change "statistically insignificant" to "not statistically significant"

On page 3., l. 7: you have changed "low temperatures" to "cold temperatures"; I would suggest the opposite. It is not that important (and "cold temperatures" is frequently used), but really only air can be cold not temperatures.

Correct this citation, the correct spelling is: Wegner, T., Grooß, J.-U., von Hobe, M., Stroh, F., Sumińska-Ebersoldt, O., Volk, C. M., Hösen, E., Mitev, V., Shur, G., and Müller, R.: Heterogeneous chlorine activation on stratospheric aerosols and clouds in the Arctic polar vortex, Atmos. Chem. Phys., 12, 11095-11106, doi:10.5194/acp-12-11095-2012, 2012.

---

## Author Response (AR2)

**RESPONSE TO THE REVIREW OF THE REVISED VERSION OF THE MANUSCRIPT**

The authors have done a very through job revising their paper. They have taken great care of taking all the comments by all the reviewers into account. I have a few remaining comments, partly also on newly added material that I would suggest the authors take into account when revising their paper for publication in ACP.

The authors have added an analysis of the halogen induced ozone loss in the model vs VPSC (new Fig. 5). I think this is a valuable addition to the paper. The authors obtain different slopes of VPSC against chemical ozone loss for different time periods due to changes in the stratospheric halogen loading (as expected). There have been attempts to construct a variant of VPSC (PACL, Tilmes et al., 2007) than includes the changing halogen loading of the stratosphere in the definition. This quantity has been applied to an evaluation of heterogeneous processes in the polar lower stratosphere in the Whole Atmosphere Community Climate Model (WACCM3) for the period 1960—2003. It would be interesting to see the behaviour of the simulated ozone loss against PACL and see if the different slopes would collapse onto one curve. It would be good to add this aspect to the paper.

Unfortunately, we are unable to calculate the PACL diagnostic as defined in Tilmes et al. (2007) due to the lack of daily mean data from six of the ensemble members (required to calculate the meteorological part of the PACL diagnostic). However, a similar figure to Fig. 5 of our manuscript using the same Vpsc diagnostic, as defined in our manuscript, normalised by multiplying with November to March mean 65-90°N 20 km $Cl_y$ and dividing by the maximum 65-90°N 20 km $Cl_y$ in a given ensemble member is shown below. As evident in the figure, the gradients of the linear fits characterising the first two and the last two periods in the 2001-2080 interval are now, as expected, similar within the associated uncertainty. However, there are differences between Figure R1 here and Figure 17 in Tilmes et al. (2007) that could result from a number of factors (including the use of meridional averages instead of vortex averages) and requires further investigation. Therefore, we prefer not to include this aspect to the final version of the manuscript.

[Figure]

Figure R1. Scatterplot for November to March mean potential $V_{PSC}$ normalised by multiplying with November to March mean 65-90°N 20 km $Cl_y$ and dividing by the maximum 65-90°N 20 km $Cl_y$ in a given ensemble member [×10^6 km^3] against halogen induced ozone loss [DU] over 1-25 km integrated over the same months. Colours, lines and a red star as in Fig. 5 in the manuscript.

There was also a lot of discussion in the reviews about the representation of polar chemistry in general and heterogeneous processes in particular in the UM-UKCA model. The authors have explained now their equilibrium NAT scheme, where NAT is formed at the NAT equilibrium temperature, which I think is acceptable for this paper. However, it is known since a long time that NAT does not form at equilibrium so I would argue it should no longer be a "common technique employed in CCMs".

We have now added a sentence to the manuscript that says: "We note that this is a simple approach; in reality the formation of NAT particles requires supersaturation of $HNO_3$ over NAT to occur (see e.g. Solomon et al., 2015)."

An aspect that I did not

mention in the first review is the parametrization for reactivity on NAT that is used in the model. There are two competing and rather different formulations for NAT reactivity (Carslaw et al., 1997a,b); as NAT is important in the heterogeneous chemistry used here it might be worth clarifying this point in the paper.

We have now added a table to the supplement (Table S1) that gives the information about the heterogeneous reactions on NAT/ICE PSCs in the model, and the parametrisation for their reactive uptake coefficients.

Further, the heterogeneous reaction HCl + ClONO2 is not included on liquid surfaces. But it is well known to occur on liquid surfaces (e.g., Hanson et al., 1994), so what is the reason for not including it? Possibly, the effects of a too early onset of NAT chemistry and the neglect of HCl + ClONO2 on liquid surfaces cancel out to some extent. There is no point in changing anything here with regard to the manuscript in question. However, I suggest correcting these two issues in future model versions, i.e. include HCl + ClONO2 on liquid surfaces and introduce a supersaturation requirement for NAT formation.

We agree that the two issues should be corrected in the future; also attempts to include the HCl + ClONO$_2$ reaction on sulphate aerosols in the model have recently been made (J.M. Keeble, pers. comm.).

The details of the heterogeneous chemistry could however matter for predictions of the future; for example it is stated in the paper now that :"Stratospheric H2O and HNO3 levels are projected to increase in the future, which is likely to enhance levels of PSCs" However it depends on the type of PSC in question in how far an increase of H2O or HNO3 will affect chlorine activation. Nonetheless, the halogen induced polar ozone loss in the model will be in many respects robust against assumptions on heterogeneous chemistry (e.g. NAT vs. liquid; Solomon et al., 2015, see also discussion and references in my previous review). As this aspect is relevant for the model results presented here, I suggest discussing the issue briefly in the paper. For example on p 5, l. 5 you could briefly state not only that Keeble et al. (2014) obtained a reasonable representation of springtime Antarctic ozone, but also why. Perhaps something along these lines ~ "Previous studies (refs) obtain a consistent and relatively realistic representation of springtime Antarctic ozone for rather different assumptions on PSC formation so that the details of the heterogeneous chemistry in the model runs presented here should have no strong impact on the simulated halogen induced ozone loss. Moreover, we compare here the time evolution of polar ozone in a consistent framework, so that possible model biases should be less important". Just a suggestion.

Since confident assessment of the relative importance of particular details of the heterogeneous chemistry scheme for the simulated Arctic chemical ozone loss in the model would require further assessment, we prefer to refrain from discussing this at this stage in the paper.

Finally, an important point in the reviews was also the discussion of chemical losses inside and outside the polar vortex. And a comparison between the 2060 and the 2063 model case. I see that the authors have responded in detail to this point and have modified and improved the manuscript accordingly. I think the analysis conducted for this discussion would also be very helpful to the reader (and not only to the reviewer/editor) and should be added to the manuscript. Perhaps in the supplement, but I suggest adding table 1 and Figs. R4 and R5 (or at least the left-hand columns of R4 and R5) to the actual paper together with the accompanying discussion from the reply (which then needs perhaps be a bit extended/reformulated). But the material is there and I would recommend showing it.

As suggested, we have added Table 1 and Fig. R4-R5 from our general response to the supplement (now called Table S2 and Fig. S4-S5, respectively)

In summary, I think the authors have done a very careful job in replying to the reviews and in revising the paper. I believe some further improvements are possible as outlined above. With these modifications, which should not be too difficult to do, I recommend the paper for publication in ACP.

Technical issues

Throughout the paper: change "statistically insignificant" to "not statistically significant"

Corrected.

On page 3., l. 7: you have changed "low temperatures" to "cold temperatures"; I would suggest the opposite. It is not that important (and "cold temperatures" is frequently used), but really only air can be cold not temperatures.

Corrected.

[revised manuscript text omitted]

**Supplementary Tables and Figures**

| Reaction | NAT PSCs | ICE PSCs |
|---|---|---|
| $HCl + ClONO_2 \rightarrow 2{\times}Cl + HNO_3$ | $\gamma = 0.3$ | $\gamma = 0.3$ |
| $ClONO_2 + H_2O \rightarrow HOCl + HNO_3$ | $\gamma = 0.006$ | $\gamma = 0.3$ |
| $N_2O_5 + H_2O \rightarrow 2{\times}HNO_3$ | $\gamma = 0.0006$ | $\gamma = 0.03$ |
| $N_2O_5 + HCl \rightarrow Cl + NO_2 + HNO_3$ | $\gamma = 0.003$ | $\gamma = 0.03$ |
| $HOCl + HCl \rightarrow H_2O + 2{\times}Cl$ | $\gamma = 0.3$ | $\gamma = 0.3$ |

Table S1. Heterogeneous chemical reactions on NAT and ICE PSCs in the model and their reactive uptake coefficients, $\gamma$.

| | 65-90°N | $PV_{850K} \geq 600$ PVU | $PV_{450K} \geq 30$ PVU | $PV_{450K} \geq 35$ PVU |
|---|---|---|---|---|
| 2063 | 39 | 44 | 46 | 43 |
| 2060 | 18 | 23 | 23 | 6 |

Table S2. Cumulative halogen induced ozone loss [DU] (1 Nov-30 Mar, 1-25 km) for the two case study model years 2063 and 2060, calculated for the 65-90°N mean and a number of polar vortex edge definitions.

[Figure]

[Figure]

Figure S1. The evolution of total ozone column [DU] over 1979-2012 for 65-90°N March (top) and over 1979-2011 for 65-90°S October (bottom) in the nudged UMUKCA CCMI REFC1 CheS+(SD) integration (red) and observations (black, Bodeker total ozone column dataset: Bodeker et al., 2005; Müller et al., 2008). See Sect. 2.1 for details.

[Figure]

Figure S2. (a-g) Timeseries of 65-90°N March total column ozone [DU] for individual ensemble members as labelled (black). Red lines show the corresponding linear trends over the 2000-2080 period.

[Figure]

Figure S3. Timeseries of 65-90°N daily mean ClO and $Cl_2O_2$ [ppt] (left) and HCl and $ClONO_2$ [ppb] (right) at 21.5 km for the model case study years 2063 (solid lines) and 2060 (dashed lines).

**1 March, 2060**

[Figure]

(a) ozone/DU

| | | | | | | | | | | | | |
|---|---|---|---|---|---|---|---|---|---|---|---|---|
| 300 | 320 | 340 | 360 | 380 | 400 | 420 | 440 | 460 | 480 | 500 | 520 | 540 |

(b) 21.5 km temperature/K

| | | | | | | | | | | |
|---|---|---|---|---|---|---|---|---|---|---|
| 180 | 185 | 190 | 195 | 200 | 205 | 210 | 215 | 220 | 225 | 230 |

(c) halogen loss/DU*day⁻¹

| | | | | | | | | | | | | |
|---|---|---|---|---|---|---|---|---|---|---|---|---|
| 0 | 0.1 | 0.2 | 0.3 | 0.4 | 0.5 | 0.6 | 0.7 | 0.8 | 0.9 | 1 | 1.1 | 1.2 |

(d) 21.5 km ClO/ppt

| | | | | | | | | | | |
|---|---|---|---|---|---|---|---|---|---|---|
| 0 | 50 | 100 | 150 | 200 | 250 | 300 | 350 | 400 | 450 | 500 | 550 |

Figure S4. Daily mean total ozone column [DU] (a), temperature at 21.5 km [K] (b), halogen induced ozone loss in the 1-25 km layer [DU/day] (c) and ClO [ppt] at 21.5 km (d) simulated on 1 March in the case study model year 2060.

1 March, 2063

(a) ozone/DU

300 320 340 360 380 400 420 440 460 480 500 520 540

(b) 21.5 km temperature/K

180 185 190 195 200 205 210 215 220 225 230

(c) halogen loss/DU*day⁻¹

0 0.1 0.2 0.3 0.4 0.5 0.6 0.7 0.8 0.9 1 1.1 1.2

(d) 21.5 km ClO/ppt

0 50 100 150 200 250 300 350 400 450 500 550

Figure S5. As in Fig. S4 but for the case study model year 2063.

As noted in Sect. 2.2 of the manuscript, our UMUKCA REFC2 ensemble of integrations consists of 2 full 1960-2099 integrations (ENS1-2) and 5 shorter runs covering November 1980 to December 2080 (ENS3-7). For technical reasons, data from 5 six-year-long intervals were excluded from the analysis, in particular:

-           July 2025 – June 2031 in member number 5 (ENS5)

5 -          April 2074 – March 2080 in member number 5 (ENS5)

-           April 1996 – March 2002 in member number 6 (ENS6)

-           April 2043 – March 2049 in member number 6 (ENS6)

-           August 1982 – July 1988 in member number 7 (ENS7)

An example of the resulting timeseries is shown in Fig. S2 for 65-90ºN March total ozone column.

**Supplementary Tables and Figures**

| Reaction | NAT PSCs | ICE PSCs |
|---|---|---|
| $HCl + ClONO_2 \rightarrow 2 \times Cl + HNO_3$ | $\gamma = 0.3$ | $\gamma = 0.3$ |
| $ClONO_2 + H_2O \rightarrow HOCl + HNO_3$ | $\gamma = 0.006$ | $\gamma = 0.3$ |
| $N_2O_5 + H_2O \rightarrow 2 \times HNO_3$ | $\gamma = 0.0006$ | $\gamma = 0.03$ |
| $N_2O_5 + HCl \rightarrow Cl + NO_2 + HNO_3$ | $\gamma = 0.003$ | $\gamma = 0.03$ |
| $HOCl + HCl \rightarrow H_2O + 2 \times Cl$ | $\gamma = 0.3$ | $\gamma = 0.3$ |

Table S1. Heterogeneous chemical reactions on NAT and ICE PSCs in the model and their reactive uptake coefficients, $\gamma$.

| | 65-90°N | $PV_{850K} \geq 600$ PVU | $PV_{450K} \geq 30$ PVU | $PV_{450K} \geq 35$ PVU |
|---|---|---|---|---|
| **2063** | 39 | 44 | 46 | 43 |
| **2060** | 18 | 23 | 23 | 6 |

Table S2. Cumulative halogen induced ozone loss [DU] (1 Nov-30 Mar, 1-25 km) for the two case study model years 2063 and 2060, calculated for the 65-90°N mean and a number of polar vortex edge definitions.

[Figure]

[Figure]

Figure S1. The evolution of total ozone column [DU] over 1979-2012 for 65-90°N March (top) and over 1979-2011 for 65-90°S October (bottom) in the nudged UMUKCA CCMI REFC1 CheS+(SD) integration (red) and observations (black, Bodeker total ozone column dataset: Bodeker et al., 2005; Müller et al., 2008). See Sect. 2.1 for details.

[Figure]

Figure S2. (a-g) Timeseries of 65-90°N March total column ozone [DU] for individual ensemble members as labelled (black). Red lines show the corresponding linear trends over the 2000-2080 period.

[Figure]

Figure S3. Timeseries of 65-90°N daily mean ClO and $Cl_2O_2$ [ppt] (left) and HCl and $ClONO_2$ [ppb] (right) at 21.5 km for the model case study years 2063 (solid lines) and 2060 (dashed lines).

[Figure]

Figure S4. Daily mean total ozone column [DU] (a), temperature at 21.5 km [K] (b), halogen induced ozone loss in the 1-25 km layer [DU/day] (c) and ClO [ppt] at 21.5 km (d) simulated on 1 March in the case study model year 2060.

1 March, 2063

(a) ozone/DU

(b) 21.5 km temperature/K

(c) halogen loss/DU*day$^{-1}$

(d) 21.5 km ClO/ppt

Figure S5. As in Fig. S4 but for the case study model year 2063.